# Hidden Markov Transformer for Simultaneous Machine Translation

**Shaolei Zhang** [1,2]**, Yang Feng** [1,2*]

[1]Key Laboratory of Intelligent Information Processing
 Institute of Computing Technology, Chinese Academy of Sciences (ICT/CAS)
[2]University of Chinese Academy of Sciences, Beijing, China
{zhangshaolei20z,fengyang}@ict.ac.cn

## Abstract

Simultaneous machine translation (SiMT) outputs the target sequence while receiving the source sequence, and hence learning when to start translating each target token is the core challenge for SiMT task. However, it is non-trivial to learn the optimal moment among many possible moments of starting translating, as the moments of starting translating always hide inside the model and can only be supervised with the observed target sequence. In this paper, we propose a *Hidden Markov Transformer* (*HMT*), which treats the moments of starting translating as hidden events and the target sequence as the corresponding observed events, thereby organizing them as a hidden Markov model. HMT explicitly models multiple moments of starting translating as the candidate hidden events, and then selects one to generate the target token. During training, by maximizing the marginal likelihood of the target sequence over multiple moments of starting translating, HMT learns to start translating at the moments that target tokens can be generated more accurately. Experiments on multiple SiMT benchmarks show that HMT outperforms strong baselines and achieves state-of-the-art performance[1].

## 1 Introduction

Recently, with the increase of real-time scenarios such as live broadcasting, video subtitles and conferences, simultaneous machine translation (SiMT) attracts more attention (Cho & Esipova, 2016; Gu et al., 2017; Ma et al., 2019; Arivazhagan et al., 2019), which requires the model to receive source token one by one and simultaneously generates the target tokens. For the purpose of high-quality translation under low latency, SiMT model needs to learn when to start translating each target token (Gu et al., 2017), thereby making a wise decision between waiting for the next source token (i.e., READ action) and generating a target token (i.e., WRITE action) during the translation process.

However, learning when to start translating target tokens is not trivial for a SiMT model, as the moments of starting translating always hide inside the model and we can only supervise the SiMT model with the observed target sequence (Zhang & Feng, 2022a). Existing SiMT methods are divided into fixed and adaptive in deciding when to start translating. Fixed methods directly decide when to start translating according to pre-defined rules instead of learning them (Dalvi et al., 2018; Ma et al., 2019; Elbayad et al., 2020). Such methods ignore the context and thus sometimes force the model to start translating even if the source contents are insufficient (Zheng et al., 2020a). Adaptive methods dynamically decide READ/WRITE actions, such as predicting a variable to indicate READ/WRITE action (Arivazhagan et al., 2019; Ma et al., 2020; Miao et al., 2021). However, due to the lack of clear correspondence between READ/WRITE actions and the observed target sequence (Zhang & Feng, 2022c), it is difficult to learn precise READ/WRITE actions only with the supervision of the observed target sequence (Alinejad et al., 2021; Zhang & Feng, 2022a; Indurthi et al., 2022).

To seek the optimal moment of starting translating each target token that hides inside the model, an ideal solution is to clearly correspond the moments of starting translating to the observed target

---

[*] Corresponding author: Yang Feng.
[1] Code is available at https://github.com/ictnlp/HMT

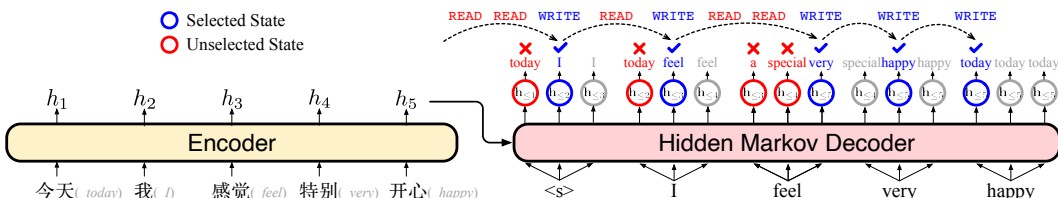

Figure 1: Illustration of hidden Markov Transformer. HMT explicitly produces $K = 3$ states for each target token to represent starting translating the target token when receiving different numbers of source tokens respectively (where $\mathbf{h}_{\leq n}$ means starting translating when receiving the first $n$ source tokens). Then, HMT judges whether to select each state from low latency to high latency (i.e., from left to right). Once a state is selected, HMT will translate the target token based on the selected state.

sequence, and further learn to start translating at those moments that target tokens can be generated more accurately. To this end, we propose *Hidden Markov Transformer* (*HMT*) for SiMT, which treats the moments of starting translating as hidden events and treats the translation results as the corresponding observed events, thereby organizing them in the form of hidden Markov model (Baum & Petrie, 1966; Rabiner & Juang, 1986; Wang et al., 2018). As illustrated in Figure 1, HMT explicitly produces a set of states for each target token, where multiple states represent starting translating the target token at different moments respectively (i.e., start translating after receiving different numbers of source tokens). Then, HMT judges whether to select each state from low latency to high latency. Once a state is selected, HMT will generate the target token based on the selected state. For example, HMT produces 3 states when generating the first target token to represent starting translating after receiving the first 1, 2 and 3 source tokens respectively (i.e., $\mathbf{h}_{\leq 1}$, $\mathbf{h}_{\leq 2}$ and $\mathbf{h}_{\leq 3}$). Then during the judgment, the first state is not selected, the second state is selected to output '*I*' and then the third state is not considered anymore, thus HMT starts translating '*I*' when receiving the first 2 source tokens. During training, HMT is optimized by maximizing the marginal likelihood of the target sequence (i.e., observed events) over all possible selection results (i.e., hidden events). In this way, those states (moments) which generate the target token more accurately will be selected more likely, thereby HMT effectively learns when to start translating under the supervision of the observed target sequence. Experiments on English→Vietnamese and German→English SiMT benchmarks show that HMT outperforms strong baselines under all latency and achieves state-of-the-art performance.

## 2 RELATED WORK

Learning when to start translating is the key to SiMT. Recent SiMT methods fall into fixed and adaptive. For fixed method, Ma et al. (2019) proposed a wait-k policy, which first READs $k$ source tokens and then READs/WRITEs one token alternately. Elbayad et al. (2020) proposed an efficient training for wait-k policy, which randomly samples different $k$ between batches. Zhang & Feng (2021a) proposed a char-level wait-k policy. Zheng et al. (2020a) proposed adaptive wait-k, which integrates multiple wait-k models heuristically during inference. Guo et al. (2022) proposed post-evaluation for the wait-k policy. Zhang et al. (2022) proposed wait-info policy to improve wait-k policy via quantifying the token information. Zhang & Feng (2021c) proposed a MoE wait-k to learn multiple wait-k policies via multiple experts. For adaptive method, Gu et al. (2017) trained an READ/WRITE agent via reinforcement learning. Zheng et al. (2019b) trained the agent with golden actions generated by rules. Zhang & Feng (2022c) proposed GMA to decide when to start translating according to the predicted alignments. Arivazhagan et al. (2019) proposed MILk, which uses a variable to indicate READ/WRITE and jointly trains the variable with monotonic attention (Raffel et al., 2017). Ma et al. (2020) proposed MMA to implement MILk on Transformer. Liu et al. (2021) proposed CAAT for SiMT to adopt RNN-T to the Transformer architecture. Miao et al. (2021) proposed GSiMT to generate READ/WRITE actions, which implicitly considers all READ/WRITE combinations during training and takes one combination in inference. Zhang & Feng (2022d) proposed an information-transport-based policy for SiMT.

Compared with the previous methods, HMT explicitly models multiple possible moments of starting translating in both training and inference, and integrates two key issues in SiMT, 'learning when to start translating' and 'learning translation', into a unified framework via the hidden Markov model.

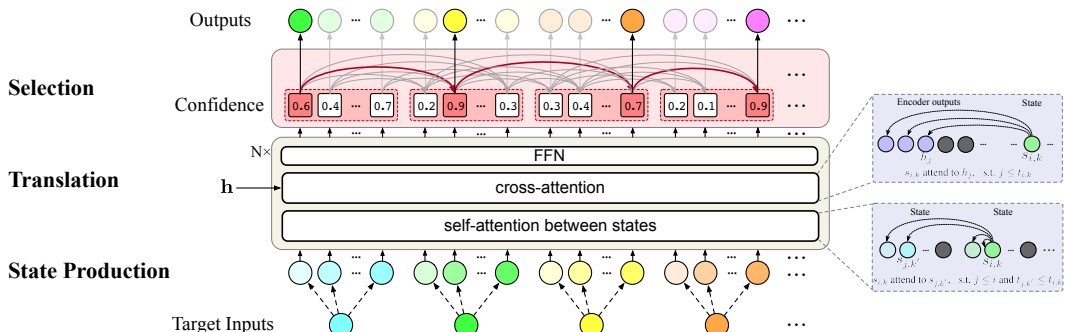

Figure 2: The architecture of hidden Markov decoder.

# 3 HIDDEN MARKOV TRANSFORMER

To learn the optimal moments of starting translating that hide inside the SiMT model, we propose hidden Markov Transformer (HMT) to organize the 'moments of starting translating' and 'observed target sequence' in the form of hidden Markov model. By maximizing the marginal likelihood of the observed target sequence over multiple possible moments of starting translating, HMT learns when to start translating. We will introduce the architecture, training and inference of HMT following.

## 3.1 ARCHITECTURE

Hidden Markov Transformer (HMT) consists of an encoder and a hidden Markov decoder. Denoting the source sequence as $\mathbf{x} = (x_1, \cdots, x_J)$ and the target sequence as $\mathbf{y} = (y_1, \cdots, y_I)$, a unidirectional encoder (Arivazhagan et al., 2019; Ma et al., 2020; Miao et al., 2021) is used to map $\mathbf{x}$ to the source hidden states $\mathbf{h} = (h_1, \cdots, h_J)$. Hidden Markov decoder explicitly produces a set of states for $y_i$ corresponding to starting translating $y_i$ at multiple moments, and then judges which state is selected to output the target token. Specifically, as shown in Figure 2, hidden Markov decoder involves three parts: state production, translation and selection.

**State Production** When translating $y_i$, HMT first produces a set of $K$ states $\mathbf{s}_i = \{s_{i,1}, \cdots, s_{i,K}\}$ to represent starting translating $y_i$ when receiving different numbers of source tokens respectively. Then, we set the *translating moments* $\mathbf{t}_i = \{t_{i,1}, \cdots, t_{i,K}\}$ for these states, where state $s_{i,k}$ is required to start translating $y_i$ when receiving the first $t_{i,k}$ source tokens $\mathbf{x}_{\leq t_{i,k}}$.

To set the suitable $\mathbf{t}$ for states, we aim to pre-prune those unfavorable translating moments, such as translating $y_1$ after receiving $x_J$ or translating $y_I$ after receiving $x_1$ (Zhang & Feng, 2022b;a). Therefore, as shown in Figure 3, we introduce a wait-$L$ path as the lower boundary and a wait-$(L+K-1)$ path as the upper boundary accordingly (Zheng et al., 2019a), and then consider those suitable moments of starting translating within this interval, where $L$ and $K$ are hyperparameters. Formally, the translating moment $t_{i,k}$ of the state $s_{i,k}$ is defined as:

$$t_{i,k} = \min\left\{L + (i-1) + (k-1), |\mathbf{x}|\right\}. \quad (1)$$

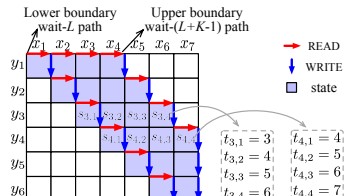

Figure 3: Setting of translating moment $\mathbf{t}$ (e.g., $L = 1$, $K = 4$).

**Translation** The representations of $K$ states for each target token are initialized via upsampling the target inputs $K$ times, and then calculated through $N$ Transformer decoder layers, each of which consists of three sub-layers: self-attention between states, cross-attention and feed-forward network.

For self-attention between states, state $s_{i,k}$ can pay attention to the state $s_{j,k'}$ of all previous target tokens (i.e., $j \leq i$), while ensuring that $s_{j,k'}$ starts translating no later than $s_{i,k}$ (i.e., $t_{j,k'} \leq t_{i,k}$) to avoid future source information leaking from $s_{j,k'}$ to $s_{i,k}$. The self-attention from $s_{i,k}$ to $s_{j,k'}$ is:

$$\text{SelfAtt}\left(s_{i,k}, s_{j,k'}\right) = \begin{cases} \text{softmax}\left(\dfrac{s_{i,k}\mathbf{W}^Q\left(s_{j,k'}\mathbf{W}^K\right)^\top}{\sqrt{d}}\right) & \text{if } j \leq i \text{ and } t_{j,k'} \leq t_{i,k} \\ 0 & \text{otherwise} \end{cases}, \quad (2)$$

where $\mathbf{W}^Q$ and $\mathbf{W}^K$ are learnable parameters. Owing to the self-attention between states, HMT can capture more comprehensive state representations by considering different moments of starting translating. For cross-attention, since state $s_{i,k}$ starts translating when receiving the first $t_{i,k}$ source tokens, $s_{i,k}$ can only focus on the source hidden state $h_j$ with $j \leq t_{i,k}$, calculated as:

$$\text{CrossAtt}\left(s_{i,k}, h_j\right) = \begin{cases} \text{softmax}\left(\frac{s_{i,k}\mathbf{W}^Q\left(h_j\mathbf{W}^K\right)^\top}{\sqrt{d}}\right) & \text{if } j \leq t_{i,k} \\ 0 & \text{otherwise} \end{cases}. \tag{3}$$

Through $N$ decoder layers, we get the final representation of state $s_{i,k}$. Accordingly, the translation probability of $y_i$ from the state $s_{i,k}$ is calculated based on its final representation:

$$p\left(y_i \mid \mathbf{x}_{\leq t_{i,k}}, \mathbf{y}_{<i}\right) = \text{softmax}\left(s_{i,k}\mathbf{W}^O\right), \tag{4}$$

where $\mathbf{W}^O$ are learnable parameters and $\mathbf{y}_{<i}$ are the previous target tokens.

**Selection** After getting the final representations of states, in order to judge whether to select state $s_{i,k}$ to generate $y_i$, HMT predicts a confidence $c_{i,k}$ of selecting state $s_{i,k}$. The confidence $c_{i,k}$ is predicted based on the final state representation of $s_{i,k}$ and the received source contents:

$$c_{i,k} = \text{sigmoid}\left(\left[\overline{\mathbf{h}}_{\leq t_{i,k}} : s_{i,k}\right]\mathbf{W}^S\right), \tag{5}$$

where $\overline{\mathbf{h}}_{\leq t_{i,k}} = \frac{1}{t_{i,k}}\sum_{j=1}^{t_{i,k}} h_j$ is the average pooling result on the hidden states of the received source tokens, $[:]$ is concatenating operation and $\mathbf{W}^S$ are learnable parameters. In inference, HMT judges whether to select the state from $s_{i,1}$ to $s_{i,K}$. If $c_{i,k} \geq 0.5$ (i.e., achieving enough confidence), HMT selects the state $s_{i,k}$ to generate $y_i$. Otherwise HMT moves to the next state $s_{i,k+1}$ and repeats the judgment. Note that we set $c_{i,K} = 1$ to ensure that HMT starts translating before the last state $s_{i,K}$.

### 3.2 TRAINING

Since when to start translating is hidden inside the model while the target sequence is observable, HMT treats when to start translating target tokens (i.e., which states are selected) as hidden events and treats target tokens as the observed events. Further, HMT organizes them in the form of hidden Markov model, thereby associating the moment of starting translating with the observed target token. Formally, for hidden events, we denoted which states are selected as $\mathbf{z} = (z_1, \cdots, z_I)$, where $z_i \in [1, K]$ represents selecting state $s_{i,z_i}$ to generate $y_i$. Then, following the HMM form, we introduce the *transition probability* between selections and the *emission probability* from the selection result.

*Transition probability* expresses the probability of the selection $z_i$ conditioned on the previous selection $z_{i-1}$, denoted as $p(z_i \mid z_{i-1})$. Since HMT judges whether to select states from $s_{i,1}$ to $s_{i,K}$ (i.e., from low latency to high latency), $s_{i,k+1}$ can be selected only if the previous state $s_{i,k}$ is not selected. Therefore, the calculation of $p(z_i \mid z_{i-1})$ consists of two parts[2]: (1) $s_{i,z_i}$ is confident to be selected, and (2) those states whose translating moment between $t_{i-1,z_{i-1}}$ and $t_{i,z_i}$ (i.e., those states that were judged before $s_{i,z_i}$) are not confident to be selected, calculated as[3]:

$$p(z_i \mid z_{i-1}) = \begin{cases} c_{i,z_i} \times \prod\limits_{\substack{l \\ t_{i-1,z_{i-1}} \leq t_{i,l} < t_{i,z_i}}} (1 - c_{i,l}) & \text{if } t_{i,z_i} \geq t_{i-1,z_{i-1}} \\ 0 & \text{if } t_{i,z_i} < t_{i-1,z_{i-1}} \end{cases}. \tag{6}$$

*Emission probability* expresses the probability of the observed target token $y_i$ from the selected state $s_{i,z_i}$, i.e., the translation probability in Eq.(4). For clarity, we rewrite it as $p\left(y_i \mid \mathbf{x}_{\leq t_{i,z_i}}, \mathbf{y}_{<i}, z_i\right)$ to emphasize the probability of HMT generating the target token $y_i$ under the selection $z_i$.

**HMM Loss** To learn when to start translating, we train HMT by maximizing the marginal likelihood of the target sequence (i.e., observed events) over all possible selection results (i.e., hidden events), thereby HMT will give higher confidence to selecting those states that can generate the target token more accurately. Given the transition probabilities and emission probabilities, the marginal likelihood of observed target sequence $\mathbf{y}$ over all possible selection results $\mathbf{z}$ is calculated as:

$$p(\mathbf{y} \mid \mathbf{x}) = \sum_{\mathbf{z}} p(\mathbf{y} \mid \mathbf{x}, \mathbf{z}) \times p(\mathbf{z}) = \sum_{\mathbf{z}} \prod_{i=1}^{|\mathbf{y}|} p\left(y_i \mid \mathbf{x}_{\leq t_{i,z_i}}, \mathbf{y}_{<i}, z_i\right) \times p(z_i \mid z_{i-1}). \tag{7}$$

---

[2]Please refer to Appendix A for detailed instructions of the transition probability between selections.

[3]We add a selection $z_0$ with $p(z_0) = 1$ before $z_1$ to indicate that no source tokens are received at the beginning of translation, i.e., $t_{0,z_0} = 0$. Therefore, $p(z_1 \mid z_0)$ is the initial probability of the selection in HMT.

---

**Algorithm 1** Inference Policy of Hidden Markov Transformer

---

**Input:** Source sequence $\mathbf{x}$; Translating moments $\mathbf{t}$; Number of states $K$.
**Output:** Translated sequence $\hat{\mathbf{y}}$.
**Init:** Received source sequence $\hat{\mathbf{x}} = [\,]$, source index $j = 0$; Translated sequence $\hat{\mathbf{y}} = [\langle \text{bos} \rangle]$, target index $i = 1$.

1: **while** $\hat{y}_{i-1} \neq \langle \text{eos} \rangle$ **do**
2:     **if** $|\hat{\mathbf{x}}| < t_{i,1}$ **then**                                                   ▷ To reach the low boundary
3:         Wait for source tokens until $|\hat{\mathbf{x}}| = t_{i,1}$;     $j \leftarrow t_{i,1}$;
4:     **for** $k \leftarrow 1$ to $K$ **do**
5:         **if** $|\hat{\mathbf{x}}| > t_{i,k}$ **then continue**;         ▷ Skip $s_{i,k}$ if its translating moment $t_{i,k}$ is less than $|\hat{\mathbf{x}}|$
6:         Calculate representation of state $s_{i,k}$ and its confidence $c_{i,k}$ according to Eq.(5);
7:         **if** $c_{i,k} \geq 0.5$ **then**                                      ▷ Select ⇒ `WRITE`
8:            Translate target token $\hat{y}_i$ according to Eq.(4);
9:            $\hat{\mathbf{y}} \leftarrow \hat{\mathbf{y}} + \hat{y}_i$;    $i \leftarrow i + 1$;
10:           **Break**;
11:         **else**                                                   ▷ Unselect ⇒ `READ`
12:           Wait for the next source token $x_{j+1}$;
13:           $\hat{\mathbf{x}} \leftarrow \hat{\mathbf{x}} + x_{j+1}$;    $j \leftarrow j + 1$;
14: **return** $\hat{\mathbf{y}}$;

---

We employ dynamic programming to reduce the computational complexity of marginalizing all possible selection results, and detailed calculations refer to Appendix B. Then, HMT is optimized with the negative log-likelihood loss $\mathcal{L}_{hmm}$:

$$\mathcal{L}_{hmm} = -\log p(\mathbf{y} \mid \mathbf{x}). \tag{8}$$

**Latency Loss** Besides, we also introduce a latency loss $\mathcal{L}_{latency}$ to trade off between translation quality and latency. $\mathcal{L}_{latency}$ is also calculated by marginalizing all possible selection results $\mathbf{z}$:

$$\mathcal{L}_{latency} = \sum_{\mathbf{z}} p(\mathbf{z}) \times \mathcal{C}(\mathbf{z}) = \sum_{\mathbf{z}} \prod_{i=1}^{|\mathbf{y}|} p(z_i \mid z_{i-1}) \times \mathcal{C}(\mathbf{z}), \tag{9}$$

where $\mathcal{C}(\mathbf{z})$ is a function to measure the latency of a given selection results $\mathbf{z}$, calculated by the average lagging (Ma et al., 2019) relative to the lower boundary: $\mathcal{C}(\mathbf{z}) = \frac{1}{|\mathbf{z}|} \sum_{i=1}^{|\mathbf{z}|} (t_{i,z_i} - t_{i,1})$.

**State Loss** We additionally introduce a state loss $\mathcal{L}_{state}$ to encourage HMT to generate the correct target token $y_i$ no matter which state $z_i$ is selected (i.e., no matter when to start translating), thereby improving the robustness on the selection. Since $K$ states are fed into the hidden Markov decoder in parallel during training, $\mathcal{L}_{state}$ is calculated through a cross-entropy loss on all states:

$$\mathcal{L}_{state} = -\frac{1}{K} \sum_{i=1}^{|\mathbf{y}|} \sum_{z_i=1}^{K} \log p\big(y_i \mid \mathbf{x}_{\leq t_{i,z_i}}, \mathbf{y}_{<i}, z_i\big). \tag{10}$$

Finally, the total loss $\mathcal{L}$ of HMT is calculated as:

$$\mathcal{L} = \mathcal{L}_{hmm} + \lambda_{latency} \mathcal{L}_{latency} + \lambda_{state} \mathcal{L}_{state}, \tag{11}$$

where we set $\lambda_{latency} = 1$ and $\lambda_{state} = 1$ in our experiments.

### 3.3 INFERENCE

In inference, HMT judges whether to select each state from low latency to high latency based on the confidence, and once a state is selected, HMT generates the target token based on the state representation. Specifically, denoting the current received source tokens as $\hat{\mathbf{x}}$, the inference policy of HMT is shown in Algorithm 1. When translating $\hat{y}_i$, HMT judges whether to select the state from $s_{i,1}$ to $s_{i,K}$, so HMT will wait at least $t_{i,1}$ source tokens to reach the lower boundary (line 2). During judging, if the confidence $c_{i,k} \geq 0.5$, HMT selects state $s_{i,k}$ to generate $\hat{y}_i$ (i.e., WRITE action) according to Eq.(4), otherwise HMT waits for the next source token (i.e., READ action) and moves to the next state $s_{i,k+1}$. We ensure that HMT starts translating $\hat{y}_i$ before the last state $s_{i,K}$ (i.e., before reaching the upper boundary) via setting $c_{i,K} = 1$. Note that due to the monotonicity of the moments to start translating in SiMT, state $s_{i,k}$ will be skipped and cannot be selected (line 5) if its translating moment $t_{i,k}$ is less than the number of received source tokens $|\hat{\mathbf{x}}|$ (i.e., the moment of translating $\hat{y}_{i-1}$), which is in line with the transition between selections in Eq.(6) during training.

## 4 EXPERIMENTS

### 4.1 DATASETS

We conduct experiments on the following datasets, which are the widely used SiMT benchmarks.

**IWSLT15[4] English→Vietnamese (En→Vi)** (133K pairs) We use TED tst2012 (1553 pairs) as the validation set and TED tst2013 (1268 pairs) as the test set. Following the previous setting (Ma et al., 2020; Zhang & Feng, 2021c), we replace the token whose frequency is less than 5 by $\langle unk \rangle$, and the vocabulary sizes of English and Vietnamese are 17K and 7.7K, respectively.
**WMT15[5] German → English (De→En)** (4.5M pairs) We use newstest2013 (3000 pairs) as the validation set and newstest2015 (2169 pairs) as the test set. BPE (Sennrich et al., 2016) is applied with 32K merge operations and the vocabulary of German and English is shared.

### 4.2 EXPERIMENTAL SETTINGS

We conduct experiments on several strong SiMT methods, described as follows.

**Full-sentence MT** (Vaswani et al., 2017) Transformer model waits for the complete source sequence and then starts translating, and we also apply uni-directional encoder for comparison.
**Wait-k** (Ma et al., 2019) Wait-k policy first waits for $k$ source tokens, and then alternately translates one target token and waits for one source token.
**Multipath Wait-k** (Elbayad et al., 2020) Multipath Wait-k trains a wait-k model via randomly sampling different $k$ between batches, and uses a fixed $k$ during inference.
**Adaptive Wait-k** (Zheng et al., 2020a) Adaptive Wait-k trains a set of wait-k models (e.g., from wait-1 to wait-13), and heuristically composites these models during inference.
**MoE Wait-k[6]** (Zhang & Feng, 2021c) Mixture-of-experts (MoE) Wait-k applies multiple experts to learn multiple wait-k policies respectively, and also uses a fixed $k$ during inference.
**MMA[7]** (Ma et al., 2020) Monotonic multi-head attention (MMA) predicts a variable to indicate READ/WRITE action, and trains this variable through monotonic attention (Raffel et al., 2017).
**GMA[8]** (Zhang & Feng, 2022c) Gaussian multi-head attention (GMA) introduces a Gaussian prior to learn the alignments in attention, and decides when to start translating based on the aligned positions.
**GSiMT** (Miao et al., 2021) Generative SiMT (GSiMT) generates a variable to indicate READ/WRITE action. GSiMT considers all combinations of READ/WRITE actions during training and only takes one combination of READ/WRITE actions in inference.
**HMT** The proposed hidden Markov Transformer, described in Sec.3.

**Settings** All systems are based on Transformer (Vaswani et al., 2017) from Fairseq Library (Ott et al., 2019). Following Ma et al. (2020) and Miao et al. (2021), we apply Transformer-Small (4 heads) for En→Vi, Transformer-Base (8 heads) and Transformer-Big (16 heads) for De→En. Due to the high training complexity, we only report GSiMT on De→En with Transformer-Base, the same as its original setting (Miao et al., 2021). The hyperparameter settings are reported in Appendix D.

**Evaluation** For SiMT performance, we report BLEU score (Papineni et al., 2002) for translation quality and Average Lagging (AL) (Ma et al., 2019) for latency. AL measures the token number that outputs lag behind the inputs. For comparison, we adjust $L$ and $K$ in HMT to get the translation quality under different latency, and the specific setting of $L$ and $K$ are reported in Appendix E

### 4.3 MAIN RESULTS

We compare HMT with the existing SiMT methods in Figure 4, where HMT outperforms the previous methods under all latency. Compared with fixed methods Wait-k, Multipath Wait-k and MoE Wait-k, HMT has obvious advantages as HMT dynamically judges when to start translating and thereby can handle complex inputs (Arivazhagan et al., 2019). Compared with adaptive methods,

---

[4]`nlp.stanford.edu/projects/nmt/`
[5]`statmt.org/wmt15/translation-task.html`
[6]`github.com/ictnlp/MoE-Waitk`
[7]`github.com/facebookresearch/fairseq/tree/main/examples/simultaneous_`
`translation`
[8]`github.com/ictnlp/GMA`

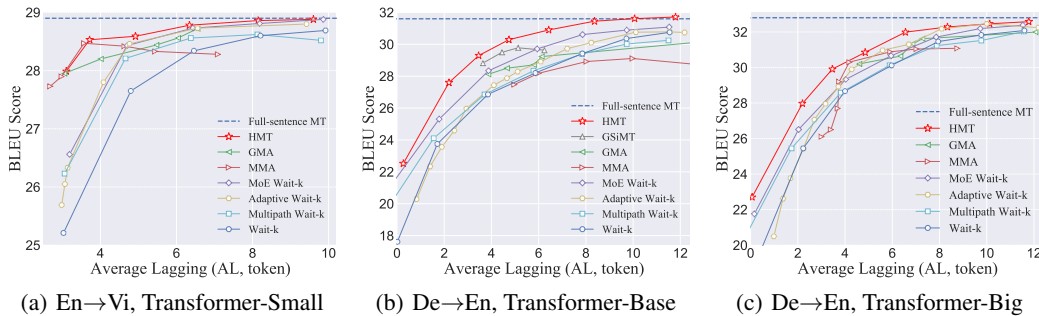

Figure 4: Translation quality (BLEU) against latency (AL) of HMT and previous SiMT methods.

HMT outperforms the current state-of-the-art MMA and GSiMT, owing to two main advantages. First, HMT models the moments of starting translating, which has stronger corresponding correlations with the observed target sequence compared with READ/WRITE action (Zhang & Feng, 2022c). Second, both MMA and GSiMT can only consider one combination of READ/WRITE actions in inference (Ma et al., 2020; Miao et al., 2021), while HMT can consider multiple moments of starting translating in both training and inference as HMT explicitly produces multiple states for different translating moments. Furthermore, the performance of MMA and GSiMT will drop at some latency (Ma et al., 2020), which is mainly because considering too many combinations of READ/WRITE actions in training may cause mutual interference (Elbayad et al., 2020; Zhang & Feng, 2021c; Wu et al., 2021). HMT pre-prunes those unfavorable moments of starting translating via the proposed boundaries, thereby achieving more stable and better performance under all latency.

## 4.4 TRAINING AND INFERENCE SPEEDS

We compare the training and inference speeds of HMT with previous methods on De→En with Transformer-Base, and the results are reported in Table 1. All speeds are evaluated on NVIDIA 3090 GPU.

**Training Speed** Compared with the fixed method Wait-k, the training speed of HMT is slightly slower as it upsamples the target sequence by $K$ times. Given the obvious performance improvements,

Table 1: Training and inference speeds of HMT.

| Systems | | #Para. | Training (s/batch) | Inference (s/token) |
|---|---|---|---|---|
| Full-sentence MT | | 60.9M | 0.204 | 0.0097 |
| Wait-k | | 60.9M | 0.205 | 0.0108 |
| MMA | | 62.5M | 2.112 | 0.0647 |
| GSiMT | | 60.9M | 5.090 | 0.0247 |
| HMT | $L=2, K=4$ | 60.9M | 0.531 | 0.0204 |
| | $L=5, K=6$ | 60.9M | 0.730 | 0.0162 |
| | $L=9, K=8$ | 60.9M | 1.051 | 0.0142 |

we argue that the slightly slower training speed than the fixed method is completely acceptable. Compared with adaptive methods MMA and GSiMT that compute the representation of the variable to indicate READ/WRITE circularly (Ma et al., 2020; Miao et al., 2021), the training speed of HMT has obvious advantages owing to computing representations of multiple states in parallel. Besides, compared to GSiMT considering all possible READ/WRITE combinations, the proposed lower and upper boundaries of the translating moments also effectively speed up HMT training.

**Inference Speed** Compared with MMA which adds more extra parameters to predict READ/WRITE action in each attention head (Ma et al., 2020), HMT only requires few extra parameters ($\mathbf{W}^S$ in Eq.(5)) and thereby achieves faster inference speed. Compared with GSiMT, which needs to calculate the target representation and generate READ/WRITE each at each step (Miao et al., 2021), HMT pre-prunes those unfavorable translating moments and only judges among the rest valuable moments, thereby improving the inference speed. Note that as the lower boundary $L$ increases, HMT can prune more candidate translating moments and thus make the inference much faster.

## 5 ANALYSIS

We conducted extensive analyses to study the specific improvements of HMT. Unless otherwise specified, all the results are reported on De→En test set with Transformer-Base.

Table 2: Ablation study of HMM loss, including marginalizing all possible selections or only optimizing the most probable selection result.

|  | AL | BLEU |
|---|---|---|
| Marginalizing all possible selections | **3.46** | **29.29** |
| Optimizing most probable selection | 1.77 | 24.02 |

Table 3: Ablation study of the weight $\lambda_{latency}$ of latency loss $\mathcal{L}_{latency}$.

| $\lambda_{latency}$ | AL | BLEU |
|---|---|---|
| 0.0 | 6.66 | 30.07 |
| 0.5 | 4.12 | 29.80 |
| 1.0 | **3.46** | **29.29** |
| 1.5 | 2.10 | 26.65 |
| 2.0 | 2.00 | 25.94 |

Table 4: Ablation study of the weight $\lambda_{state}$ of state loss $\mathcal{L}_{state}$.

| $\lambda_{state}$ | AL | BLEU |
|---|---|---|
| 0.0 | 3.24 | 27.74 |
| 0.5 | 3.47 | 29.10 |
| 1.0 | **3.46** | **29.29** |
| 1.5 | 3.46 | 29.16 |
| 2.0 | 3.58 | 29.21 |

## 5.1 ABLATION STUDY

We conduct multiple ablation studies, where $L=3$ and $K=6$ are applied in all ablation studies.

**HMM Loss** HMT learns when to start translating by marginalizing all possible selection results during training. To verify its effectiveness, we compare the performance of marginalizing all possible selections and only optimizing the most probable selection in Table 2, where the latter is realized by replacing $\sum_{\mathbf{z}}$ in Eq.(7) with $\text{Max}_{\mathbf{z}}$. Under the same setting of $L=3$ and $K=6$, only optimizing the most probable selection makes the model fall into a local optimum (Miao et al., 2021) of always selecting the first state, resulting in latency and translation quality close to wait-3 policy. Marginalizing all possible selections effectively enables HMT to learn when to start translating from multiple possible moments, achieving a better trade-off between translation quality and latency.

**Weight of Latency Loss** Table 3 reports the performance of HMT on various $\lambda_{latency}$. The setting of $\lambda_{latency}$ affects the trade-off between latency and translation quality. Too large (i.e., 2.0) or too small (i.e., 0.0) $\lambda_{latency}$ sometimes makes the model start translating when reaching the upper or lower boundary, while moderate $\lambda_{latency}$ achieves the best trade-off. Note that HMT is not sensitive to the setting of $\lambda_{latency}$, either $\lambda_{latency}=0.5$ or $\lambda_{latency}=1.0$ can achieve the best trade-off, where the result with $\lambda_{latency}=0.5$ is almost on the HMT line in Figure 4(b).

**Weight of State Loss** Table 4 demonstrates the effectiveness of the introduced state loss $\mathcal{L}_{state}$. $\mathcal{L}_{state}$ encourages each state to generate the correct target token, which can bring about 1.5 BLEU improvements. In addition, the translation quality of HMT is not sensitive to the weight $\lambda_{state}$ of state loss, and various $\lambda_{state}$ can bring similar improvements.

## 5.2 HOW MANY STATES ARE BETTER?

HMT produces a set of $K$ states for each target token to capture multiple moments of starting translating. To explore how many states are better, we report the HMT performance under various $K$ in Table 5, where we adopt different lower boundaries $L$ (refer to Eq.(1)) to get similar latency for comparison.

The results show that considering multiple states is significantly better than considering only one state ($K=1$), demonstrating that HMT finds a better moment to start translating from multiple states. For multiple states, a larger state number $K$ does not necessarily lead to better SiMT performance, and HMT exhibits specific preferences of $K$ at different latency levels. Specifically, a smaller $K$ performs better under low latency, and the best $K$ gradually increases as the latency increases. This is because translating moments with a large gap may interfere with each other during training (Elbayad

Table 5: HMT performance with various states number $K$.

|  | $L$ | $K$ | AL | BLEU |
|---|---|---|---|---|
| Low latency | 4 | 1 | 2.57 | 25.57 |
|  | 3 | 2 | 2.15 | 26.07 |
|  | **2** | **4** | **2.20** | **27.60** |
|  | 1 | 6 | 2.28 | 25.69 |
| Middle latency | 7 | 1 | 5.86 | 28.20 |
|  | 6 | 4 | 4.90 | 30.14 |
|  | **5** | **6** | **4.74** | **30.29** |
|  | 4 | 8 | 4.69 | 29.35 |
| High latency | 11 | 1 | 9.71 | 30.36 |
|  | 10 | 6 | 9.06 | 31.32 |
|  | **9** | **8** | **8.36** | **31.45** |
|  | 8 | 10 | 8.27 | 31.36 |

et al., 2020; Zhang et al., 2021; Zhang & Feng, 2021c), where the gap is more obvious at low latency. Taking $K=6$ as an example, the gap between READ 1 tokens/6 tokens (i.e., low latency) is more obvious than READ 10 tokens/15 tokens (i.e., high latency), as the latter contains more overlaps on the received source tokens. Owing to the proposed boundary for translating moments, HMT can avoid the interference via setting suitable $K$ for different latency levels. Further, compared with GSiMT directly considering arbitrary READ/WRITE combinations during training (Miao et al., 2021), HMT pre-prunes those unfavorable moments and thereby achieves better performance.

### 5.3 SUPERIORITY OF ATTENTION BETWEEN STATES

Self-attention between states (in Eq.(2)) enables HMT to consider multiple moments of starting translating and thereby capture comprehensive state representations. To verify the effectiveness of self-attention between states, we compare the performance of paying attention to multiple states or only one state of each target token. To this end, we apply three modes of self-attention, named *Multiple*, *Max* and *Selected*. 'Multiple' is the proposed self-attention in HMT that the state can attend to multi-

Table 6: Performance with various modes of self-attention in HMT.

| Training | Inference | AL | BLEU |
|---|---|---|---|
| Multiple | Multiple | **3.46** | **29.29** |
| Multiple | Max | 3.37 | 28.99 |
| Multiple | Selected | 3.42 | 28.83 |
| Max | Max | 3.30 | 28.66 |
| Max | Selected | 3.36 | 28.46 |

ple states of previous target tokens. 'Max' means that the state can only attend to one state with the maximum translating moments of each target token. 'Selected' means that the state can only attend to the selected state used to generate each target token, which is the most common inference way in the existing SiMT methods (i.e., once the SiMT model decides to start translating, subsequent translations will pay attention to the target representation resulting from this decision.) (Ma et al., 2019; 2020). Note that all modes of attention need to avoid information leakage between states, i.e., satisfying $t_{j,k'} \leq t_{i,k}$ in Eq.(2). The detailed introduction of these attention refer to Appendix C.6.

As shown in Table 6, using 'Multiple' in both training and inference achieves the best performance. Compared with 'Max' which focuses on the state with the maximum translating moment (i.e., containing most source information), 'Multiple' brings 0.65 BLEU improvements via considering multiple different translating moments (Zhang & Feng, 2021c). In inference, considering multiple states with different translating moments can effectively improve the robustness as SiMT model may find that it made a wrong decision at the previous step after receiving some new source tokens (Zheng et al., 2020b). For 'Selection', if SiMT model makes a wrong decision on when to start translating, subsequent translations will be affected as they can only focus on this selected state. Owing to the attention between states, HMT allows subsequent translations to focus on those unselected states, thereby mitigating the impact of uncertain decisions and bringing about 0.46 BLEU improvements.

### 5.4 QUALITY OF SELECTION

HMT judges whether to select state $s_{i,k}$ by predicting its confidence $c_{i,k}$. To verify the quality of the selection based on predicted confidence, we calculated the token accuracy under different confidences in Figure 5. There is an obvious correlation between confidence and token accuracy, where HMT learns higher confidence for those states that can generate the target tokens more accurately. Especially in inference, since HMT selects state $s_{i,k}$ to generate the target token when its confidence $c_{i,k} \geq 0.5$, HMT learns to start translating at the moments that can generate target tokens with more than 60% accuracy on average, thereby ensuring the translation quality.

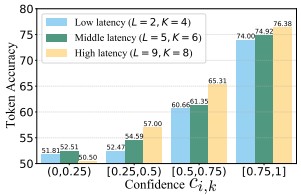

Figure 5: Token accuracy with predicted confidence.

Besides token accuracy, we further analyse the relationship between translation quality and selection result over the whole sequence. We report the BLEU score under different probabilities of selection result $p(\mathbf{z})$ in Figure 6, where we apply $p(\mathbf{z})^{\frac{1}{|\mathbf{z}|}}$ to avoid the target length $|\mathbf{z}|$ influencing $p(\mathbf{z})$. The selection results with higher probability always achieve higher BLEU scores, showing that training by marginal likelihood encourages HMT to give a higher probability to the hidden sequence (i.e., selection result over the whole sequence) that can generate the high-quality observed target sequence.

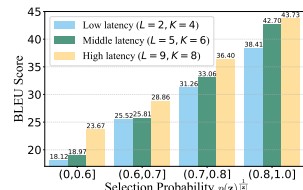

Figure 6: BLEU score with probability of selection result.

### 6 CONCLUSION

In this paper, we propose hidden Markov Transformer (HMT) for SiMT, which integrates learning when to start translating and learning translation into a unified framework. Experiments on multiple SiMT benchmarks show that HMT outperforms the strong baselines and achieves state-of-the-art performance. Further, extensive analyses demonstrate the effectiveness and superiority of HMT.

ACKNOWLEDGMENTS

We thank all the anonymous reviewers for their insightful and valuable comments.

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

# A    CALCULATION OF TRANSITION PROBABILITY

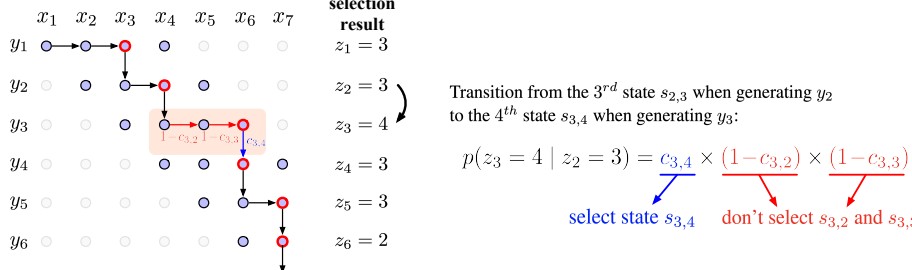

Figure 7: An example to depict the calculation of transition probability $p(z_i \mid z_{i-1})$.

We give the transition probability $p(z_i \mid z_{i-1})$ between selection results in Sec.3.2, and here we describe the calculation of transition probability in more detail.

First of all, a specific selection $z_i$ represents translating $y_i$ when receiving first $t_{i,z_i}$ source tokens through the state $s_{i,z_i}$. Similarly, selection $z_{i-1}$ represents translating $y_{i-1}$ when receiving first $t_{i-1,z_{i-1}}$ source tokens through the state $s_{i-1,z_{i-1}}$. Due to the monotonicity of the moment to start translating in SiMT, it is not possible to transfer from $z_{i-1}$ to $z_i$ if $t_{i-1,z_{i-1}} > t_{i,z_i}$. Then for $t_{i-1,z_{i-1}} \leq t_{i,z_i}$, since HMT judges whether to select each state one by one (from $s_{i,1}$ to $s_{i,K}$) and starts translating if a state is selected, the premise of the transition from $z_{i-1}$ to $z_i$ is that all states whose translating moment belongs to $[t_{i-1,z_{i-1}}, t_{i,z_i})$ are not confident to be selected, and $s_{i,z_i}$ is confident to be selected. As the example shown in Figure 7, the transition probability $p(z_3 = 4 \mid z_2 = 3)$ from $z_2$ to $z_3$ consists of the probability of unselecting $s_{3,2}$ and $s_{3,3}$, and the probability of selecting $s_{3,4}$.

Formally, the transition probability $p(z_i \mid z_{i-1})$ is calculated as:

$$p(z_i \mid z_{i-1}) = \begin{cases} c_{i,z_i} \times \prod\limits_{\substack{l \\ t_{i-1,z_{i-1}} \leq t_{i,l} < t_{i,z_i}}} (1 - c_{i,l}) & \text{if } t_{i,z_i} \geq t_{i-1,z_{i-1}} \\ 0 & \text{if } t_{i,z_i} < t_{i-1,z_{i-1}} \end{cases} . \tag{12}$$

# B    DYNAMIC PROGRAMMING IN HMT

HMT treats which states are selected (i.e, when to start translating) as hidden events and the target tokens as observed events, and organizes the generation of the target sequence as a hidden Markov model. During training, HMT is optimized by maximizing the marginal likelihood of the target sequence (i.e., observed events) over all possible selection results (i.e., hidden events):

$$p(\mathbf{y} \mid \mathbf{x}) = \sum_{\mathbf{z}} p(\mathbf{y} \mid \mathbf{x}, \mathbf{z}) \times p(\mathbf{z}) \tag{13}$$

$$= \sum_{\mathbf{z}} \prod_{i=1}^{|\mathbf{y}|} p(y_i \mid \mathbf{x}_{\leq t_{i,z_i}}, \mathbf{y}_{<i}, z_i) \times p(z_i \mid z_{i-1}). \tag{14}$$

We apply dynamic programming (a.k.a. forward algorithm in HMM) to calculate the marginal likelihood efficiently (Baum & Petrie, 1966). Formally, we introduce the intermediate variable $\alpha_i(k)$ to represent the probability of selecting the $k^{th}$ state $s_{i,k}$ when generating the first $i$ target tokens $\mathbf{y}_{\leq i}$, defined as:

$$\alpha_i(k) = p(\mathbf{y}_{\leq i}, z_i = k \mid \mathbf{x}). \tag{15}$$

**Initialization** The initial $\alpha_1(k)$ is calculated as:

$$\alpha_1(k) = \pi_k \times p(y_1 \mid \mathbf{x}_{\leq t_{1,k}}, \mathbf{y}_{<1}, z_1 = k), \tag{16}$$

where $\pi_k = p(z_1 = k)$ is the initial probability of selecting $s_{1,k}$. In the implementation, we add a certain selection $z_0$ with $p(z_0) = 1$ before $z_1$ to indicate that no source tokens are received at the

beginning of translation, i.e., $t_{0,z_0} = 0$. Accordingly, the initial probability $p(z_1)$ is calculated via the transition probability from $z_0$ to $z_1$.

**Recursion** Following, $\alpha_i(k)$ is calculated by summing over the extensions of all transitions from the previous step's selection to the current selection. Therefore, $\alpha_i(k)$ is calculated as the following recursion form:

$$\alpha_i(k) = \sum_{k'=1}^{K} \alpha_{i-1}\left(k'\right) \times p\left(z_i = k \mid z_{i-1} = k'\right) \times p\left(y_i \mid \mathbf{x}_{\leq t_{i,k}}, \mathbf{y}_{<i}, z_i = k\right). \quad (17)$$

**Termination** Finally, the marginal likelihood of the target sequence over all possible selection results is calculated as:

$$p(\mathbf{y} \mid \mathbf{x}) = \sum_{k=1}^{K} \alpha_I(k). \quad (18)$$

## C EXPANDED ANALYSES

### C.1 SPECIFIC IMPROVEMENTS OF POLICY AND TRANSLATION

The proposed HMT integrates the two key issues in SiMT 'learning when to start translating' (i.e., inference policy) and 'learning translation' (i.e., translation capability) into a unified framework. Since HMT explicitly models multiple moments of starting translating in both training and inference, HMT architecture can flexibly cooperate with other inference policies, such as wait-k policy (Ma et al., 2019), which allows us to learn the specific improvements brought by HMT architecture and inference policy. Specifically, we report the results of 'HMT architecture + Wait-k inference' in Figure 8, where wait-k inference for HMT architecture is realized by forcing HMT to always select the last state.

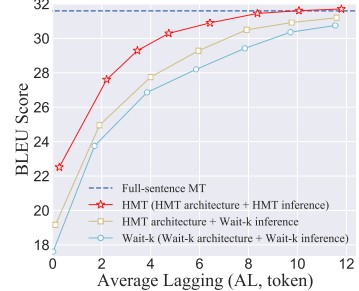

Figure 8: Specific improvements brought by HMT architecture and inference policy.

By comparison, for wait-k inference, HMT architecture has stronger translation capability due to the comprehensive consideration of multiple translating moments, thereby bringing about 0.8 BLEU improvements. Further, compared to wait-k inference, HMT inference learns more accurate moments to start translating and brings about 2.8 BLEU improvements on average. In particular, the improvements brought by the inference policy are more obvious at low latency, as the precise translating moments are more important for SiMT under low latency (Arivazhagan et al., 2019).

### C.2 ABLATION STUDY ON THRESHOLD OF CONFIDENCE

In inference, HMT will select state $s_{i,k}$ to generate $y_i$ when its confidence $c_{i,k} \geq 0.5$, where $0.5$ can be regarded as the confidence threshold that HMT believes that the state can generate the correct target token. Figure 5 also proves that the state with higher confidence can generate the target token more accurately. To study the effect of the confidence threshold in HMT, we show the HMT performance under different confidence thresholds $\delta$ in Figure 9, where HMT will select state $s_{i,k}$ to generate $y_i$ when its confidence $c_{i,k} \geq \delta$ in inference.

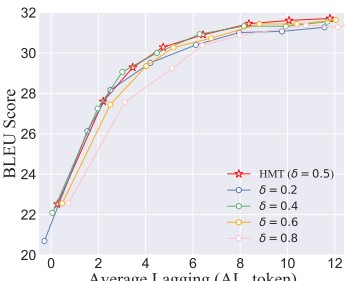

Figure 9: HMT performance under various thresholds $\delta$ of confidence.

In inference, moderate confidence thresholds, such as $\delta = 0.4$ and $\delta = 0.5$, achieve similar SiMT performance, indicating that HMT is not sensitive to the setting of the confidence threshold. Furthermore, as the confidence threshold decrease to $0.2$, HMT starts translating much earlier, resulting in a slight decrease in translation quality. As the threshold increases to $0.8$, the latency of HMT increases, but the improvement in translation quality is not obvious, which indicates that $0.5$ confidence is enough to generate the correct target token for the state in HMT.

## C.3 CASE STUDY

We conduct case studies on two difficult De→En cases in Figure 10 and Figure 11, where the word order difference between German and English is more challenging for SiMT model (i.e., the verb in German always lags behind) (Ma et al., 2019; Zhang & Feng, 2021c). We show the specific inference process of HMT and visualize the outputs of all states (including selected, unselected and not considered.) to illustrate that HMT successfully learns when to start translating.

| Source: | auf der Liste stehen 101 Pro@@ mis .  | | | |
| | *on the list stand 101 celebrities . * | | | |
| Reference: | there are 101 celeb@@ r@@ ity names on the list .  | | | |
| **Step** | **Inputs (received source sequence)** | **State** | **Confidence** | **Outputs** |
| 1 | auf der | $s_{1,1}$ | 0.09 | ~~on~~ |
| 2 | auf der Liste | $s_{1,2}$ | 0.38 | ~~on~~ |
| 3 | auf der Liste stehen | $s_{1,3}$ | 0.38 | ~~the~~ |
| 4 | auf der Liste stehen 101 | $s_{1,4}$ | 1.00 | there |
| 5 | auf der Liste stehen 101 | $s_{2,3}$ | 0.98 | are |
| 6 | auf der Liste stehen 101 | $s_{3,2}$ | 0.85 | 101 |
| 7 | auf der Liste stehen 101 | $s_{4,1}$ | 0.01 | ~~items~~ |
| 8 | auf der Liste stehen 101 Pro@@ | $s_{4,2}$ | 0.02 | ~~pro@@~~ |
| 9 | auf der Liste stehen 101 Pro@@ mis | $s_{4,3}$ | 0.65 | celeb@@ |
| 10 | auf der Liste stehen 101 Pro@@ mis | $s_{5,2}$ | 0.87 | rities |
| 11 | auf der Liste stehen 101 Pro@@ mis | $s_{6,1}$ | 0.26 | ~~in~~ |
| 12 | auf der Liste stehen 101 Pro@@ mis . | $s_{6,2}$ | 0.93 | on |
| 13 | auf der Liste stehen 101 Pro@@ mis . | $s_{7,1}$ | 0.99 | the |
| 14 | auf der Liste stehen 101 Pro@@ mis .  | $s_{8,1}$ | 1.00 | list |
| 15 | auf der Liste stehen 101 Pro@@ mis .  | $s_{9,1}$ | 1.00 | . |
| 16 | auf der Liste stehen 101 Pro@@ mis .  | $s_{10,1}$ | 1.00 |  |

(a) Inference process of HMT. 'State' records the currently considered state, and HMT selects the current state and starts translating the target token when its corresponding confidence is greater than 0.5. The outputs marked in red strikethrough represent potential outputs for those states that are not selected.

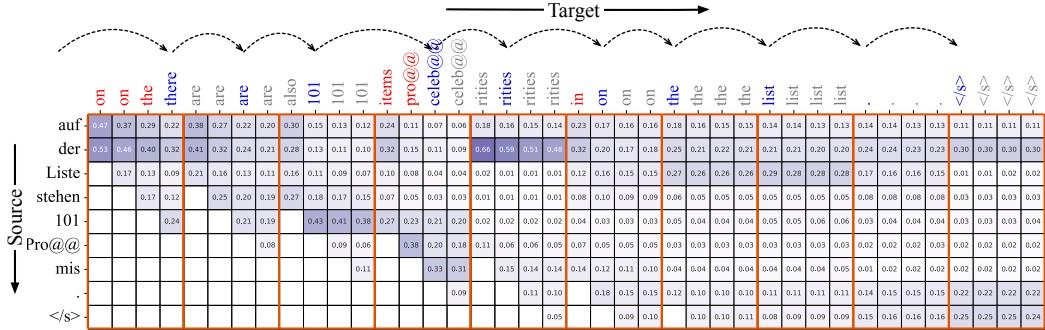

(b) Visualization of all state outputs with their cross-attention to the received source tokens. For the outputs, the outputs marked in red represent unselected states, the outputs marked in blue represent the selected states, and the outputs marked in gray represent the states that are not considered in inference (i.e., those states after the selected state, or the states whose translating moment is earlier than the moment of translating the previous target token). For the cross-attention, the numerical values in the squares report the cross-attention weight, and blank squares indicate that those source tokens have not been received when translating the target token.

Figure 10: Case study of #2124 in De→En test set, where we apply HMT with $L=2$ and $K=4$.

**Case with Word Order Difference** As shown in Figure 10, '*auf der Liste*' in German is at the beginning of the sentence, while the corresponding translation '*on the list*' is at the end in English. For this case, HMT decides not to select states $s_{1,1}$, $s_{1,2}$ and $s_{1,3}$ because their confidences are less than 0.5, especially the outputs of these states also prove that these moments are not good to start translating. Then, HMT generates '*There*' at state $s_{1,4}$. Similar situations also occur when generating '*celebrities*' and '*on*'. Figure 10(b) shows more specific outputs for all states and their cross-attention on the received source tokens, where each state corresponds to a moment of starting translating. By selecting one state from $K$ states, HMT effectively finds the optimal moment to start translating, i.e., the moment that can generate the correct target token with lower latency. In particular, those unselected states (i.e., less confidence) tend to produce incorrect translations, while the selected states can produce correct translations by paying attention to the newly received source contents. About this, we already present a statistical analysis between confidence and token accuracy

in Figure 5. Further, the selected state always tends to be the earliest state that can generate the correct translation, i.e., the state with lower latency.

| Source: | Sie schi@@ enen einen Aut@@ oun@@ fall zu erwarten , und das pas@@ sierte einfach nicht .  |||
| | *they*    *seem*    *a*    *car accident*    *to*   *expect*   *, and that*   *happened*   *simply*   *not*   *. * |||
| **Reference:** | they seemed to expect a car crash and it didn 't quite happen .  |||
| **Step** | **Inputs (received source sequence)** | **State** | **Confidence** | **Outputs** |
| 1 | Sie schi@@ enen | $s_{1,1}$ | 0.51 | they |
| 2 | Sie schi@@ enen einen | $s_{2,1}$ | 0.54 | seemed |
| 3 | Sie schi@@ enen einen Aut@@ | $s_{3,1}$ | 0.87 | to |
| 4 | Sie schi@@ enen einen Aut@@ oun@@ | $s_{4,1}$ | 0.09 | ~~be~~ |
| 5 | Sie schi@@ enen einen Aut@@ oun@@ fall | $s_{4,2}$ | 0.14 | ~~have~~ |
| 6 | Sie schi@@ enen einen Aut@@ oun@@ fall zu | $s_{4,3}$ | 0.07 | ~~have~~ |
| 7 | Sie schi@@ enen einen Aut@@ oun@@ fall zu erwarten | $s_{4,4}$ | 0.88 | expect |
| 8 | Sie schi@@ enen einen Aut@@ oun@@ fall zu erwarten | $s_{5,3}$ | 0.91 | an |
| 9 | Sie schi@@ enen einen Aut@@ oun@@ fall zu erwarten | $s_{6,2}$ | 0.94 | auto |
| 10 | Sie schi@@ enen einen Aut@@ oun@@ fall zu erwarten | $s_{7,1}$ | 0.67 | accident |
| 11 | Sie schi@@ enen einen Aut@@ oun@@ fall zu erwarten , | $s_{8,1}$ | 0.90 | , |
| 12 | Sie schi@@ enen einen Aut@@ oun@@ fall zu erwarten , und | $s_{9,1}$ | 0.64 | and |
| 13 | Sie schi@@ enen einen Aut@@ oun@@ fall zu erwarten , und das | $s_{10,1}$ | 0.55 | that |
| 14 | Sie schi@@ enen einen Aut@@ oun@@ fall zu erwarten , und das pas@@ | $s_{11,1}$ | 0.24 | ~~is~~ |
| 15 | Sie schi@@ enen einen Aut@@ oun@@ fall zu erwarten , und das pas@@ sierte | $s_{11,2}$ | 0.54 | happened |
| 16 | Sie schi@@ enen einen Aut@@ oun@@ fall zu erwarten , und das pas@@ sierte | $s_{12,1}$ | 0.06 | ~~—~~ |
| 17 | Sie schi@@ enen einen Aut@@ oun@@ fall zu erwarten , und das pas@@ sierte einfach | $s_{12,2}$ | 0.56 | simply |
| 18 | Sie schi@@ enen einen Aut@@ oun@@ fall zu erwarten , und das pas@@ sierte einfach | $s_{13,1}$ | 0.03 | ~~because~~ |
| 19 | Sie schi@@ enen einen Aut@@ oun@@ fall zu erwarten , und das pas@@ sierte einfach nicht | $s_{13,2}$ | 0.97 | not |
| 20 | Sie schi@@ enen einen Aut@@ oun@@ fall zu erwarten , und das pas@@ sierte einfach nicht | $s_{14,1}$ | 0.17 | ~~—~~ |
| 21 | Sie schi@@ enen einen Aut@@ oun@@ fall zu erwarten , und das pas@@ sierte einfach nicht . | $s_{14,2}$ | 0.95 | . |
| 22 | Sie schi@@ enen einen Aut@@ oun@@ fall zu erwarten , und das pas@@ sierte einfach nicht . | $s_{15,1}$ | 0.40 | ~~~~ |
| 23 | Sie schi@@ enen einen Aut@@ oun@@ fall zu erwarten , und das pas@@ sierte einfach nicht .  | $s_{15,2}$ | 1.00 |  |

(a) Inference process of HMT.

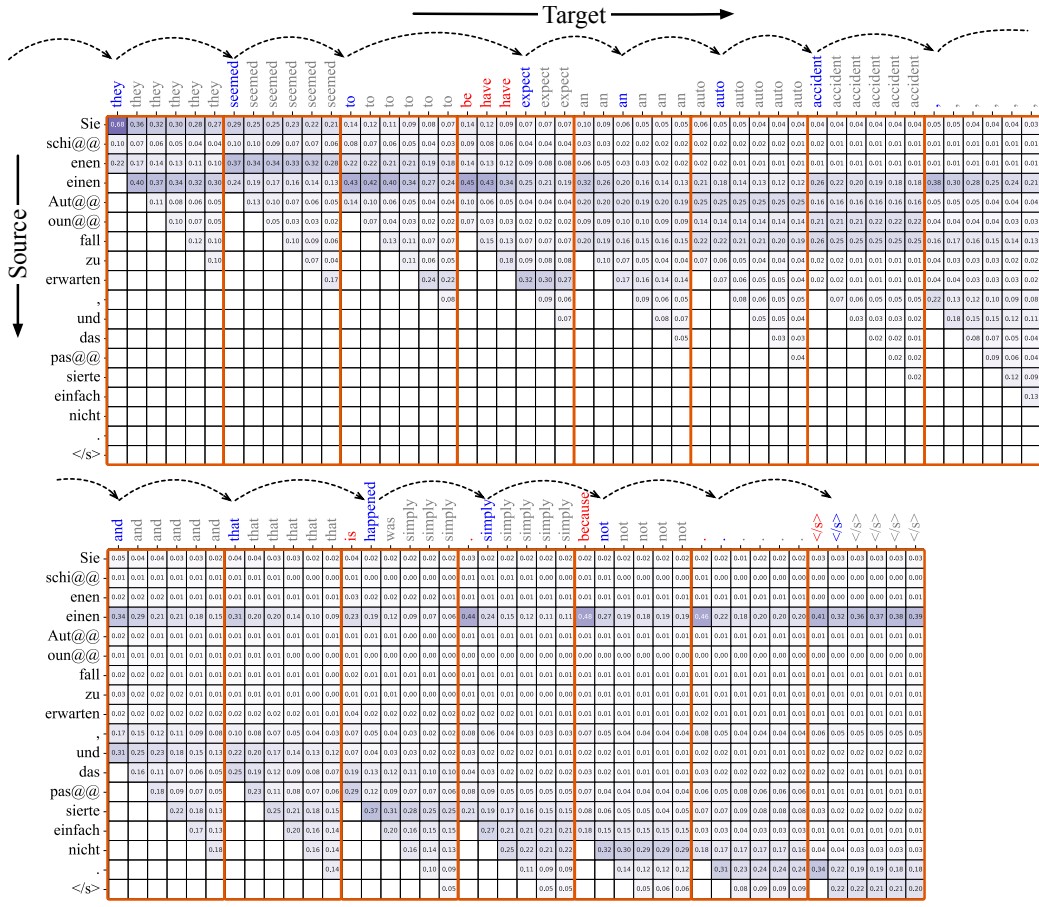

(b) Visualization of all state outputs with their cross-attention to the received source tokens.

Figure 11: Case study of #378 in De→En test set, where we apply HMT with $L = 3$ and $K = 6$.

**Case that Verb Lags Behind** Figure 11 gives a more complex case as the verb '*erwarten*' and '*passierte*' in German lag behind, which requires the SiMT model to accurately judge when to start translating (Grissom II et al., 2014; Ma et al., 2019). In HMT, when the verb has not been received, the states always get lower confidence, so HMT can start translating the corresponding '*expect*' and '*happened*' after receiving the source verb. Besides, the cross-attention in Figure 11(b) shows that HMT generates the correct translations at state $s_{4,4}$ and $s_{11,2}$ because they pay attention to the received verb '*erwarten*' and '*passierte*' (Zhang & Feng, 2021b), proving the effectiveness of predicting confidence according to the target representation and received source contents.

## C.4    EFFECT OF STATE LOSS $\mathcal{L}_{state}$

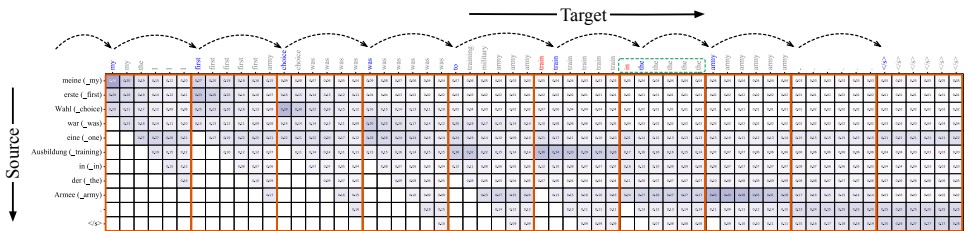

(a) HMT without the state loss $\mathcal{L}_{state}$. Translation result: '*my first choice was to train the army.*'. The green box marks the incorrect translation, where the correct translation should be '*in*' but the model generated '*the*'.

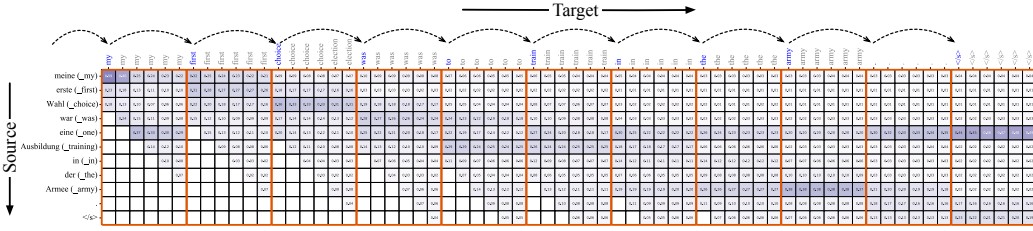

(b) HMT with the state loss $\mathcal{L}_{state}$. Translation result: '*my first choice was to train in the army.*'.

Figure 12: Effect of the proposed state loss $\mathcal{L}_{state}$. We apply HMT with $L = 3$ and $K = 6$ and visualize the state outputs of case #912 in De→En test set (Source: '*meine erste Wahl war eine Ausbildung in der Armee.*'; Reference: '*my first choice was to go in the army.*'). The outputs marked in red represent unselected states, the outputs marked in blue represent the selected states, and the outputs marked in gray represent the states that are not considered in inference. The numerical values in the squares report the cross-attention weight, and blank squares indicate that those source tokens have not been received when translating the target token.

For HMT training (refer to Sec.3.2), we propose the state loss $\mathcal{L}_{state}$ to encourage HMT to generate the correct target token no matter which state is selected (i.e., no matter when to start translating), and the ablation study in Sec.5.1 demonstrates the improvements brought by the state loss. To further study the effect of state loss $\mathcal{L}_{state}$, we visualize the state outputs of case #912 with and without $\mathcal{L}_{state}$ in Figure 12.

When removing the state loss $\mathcal{L}_{state}$ in training, those states that are selected with lower probability hardly learn to generate the correct target token, because the emission probability from these states may contribute little to the marginal likelihood of the target sequence. As shown in Figure 12(a), some later states incorrectly generate '*I*' when generating the first target token, and some states generate '*was*' when generating the third target token, etc. Although in most cases HMT will not select these states during inference, when the selection is slightly uncertain, the model may output the incorrect target token, such as generating '*the*' instead of '*in*' (marked with the green box in Figure 12(a)). As shown in Figure 12(b), with the state loss $\mathcal{L}_{state}$, multiple states in HMT all can learn to generate the correct target token, no matter when to start translating the target token. Therefore, HMT can still generate the correct target token even if it makes the wrong decision in the selection, thereby improving the robustness on the selection.

## C.5   QUALITY OF POLICY

The quality of the policy directly affects the SiMT performance, and a good policy should ensure that the model receives its corresponding source token before translating each target token, thereby achieving high-quality translation. Following Zhang & Feng (2022a), we calculate the proportion of the ground-truth aligned source tokens received before translating for the evaluation of the policy quality. We apply RWTH[9] De→En alignment dataset and perform force-decoding[10] to get the moments of translating each target token. Specifically, we denote the ground-truth aligned source position[11] of $y_i$ as $a_i$, and use $g_i$ to record the translating moments of $y_i$ (i.e., the number of received source tokens when translating $y_i$). Given the alignment $a_i$ and translating moment $g_i$, the proportion of aligned source tokens received before translating is calculated as:

$$\text{Proportion} = \frac{1}{|\mathbf{y}|} \sum_{i=1}^{|\mathbf{y}|} \mathbb{1}_{a_i \leq g_i}, \qquad (19)$$

$$\text{where} \quad \mathbb{1}_{a_i \leq g_i} = \begin{cases} 1, & a_i \leq g_i \\ 0, & a_i > g_i \end{cases}. \qquad (20)$$

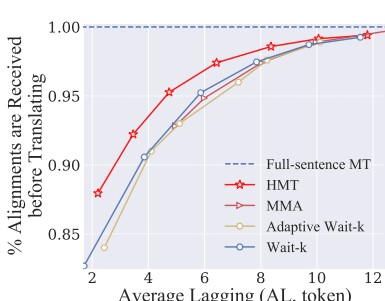

The evaluation results are shown in Figure 13. Compared with previous policies, HMT receives more aligned source tokens before translating under the same latency, which demonstrates that HMT can make more precise decisions on when to start translating. Owing to the superiority of policy, HMT can receive more aligned source tokens and thereby achieve higher translation quality than previous methods under the same latency.

Figure 13: Quality of the policy in HMT. We calculate the proportion of aligned source tokens received before translating in various policies.

## C.6   WHY SELF-ATTENTION BETWEEN STATES?

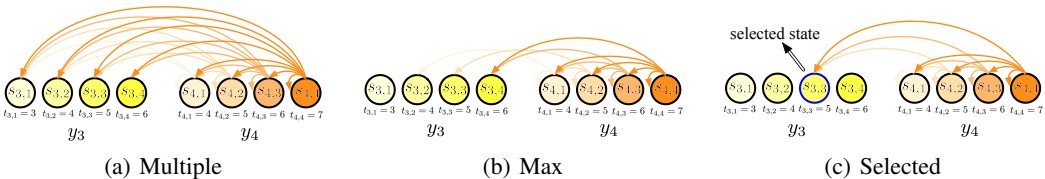

| (a) Multiple | (b) Max | (c) Selected |
|---|---|---|

Figure 14: Schematic diagram of self-attention between states in HMT, named 'Multiple'. We also propose two variants, 'Max' and 'Selected', to demonstrate the superiority of 'Multiple' attention mode. The schematic diagram shows an example of HMT with $L = 1$ and $K = 4$, where the translating moments of states for $y_3$ and $y_4$ are $\mathbf{t}_3 = (3, 4, 5, 6)$ and $\mathbf{t}_4 = (4, 5, 6, 7)$, respectively.

HMT applies self-attention among all states based on Eq.(2), and here we explain why HMT applies this attention pattern in more depth. For comparison, we introduce three modes of self-attention in Sec.5.3, named Multiple, Max and Selected:

- **Multiple**: The self-attention mode between states in HMT. In 'Multiple', the state can pay attention to multiple states of previous target tokens. Taking Figure 14(a) as an example, when translating $y_4$, state $s_{4,3}$ can pay attention to $s_{3,1}, s_{3,2}, s_{3,3}$ and $s_{3,4}$ of $y_3$, meanwhile state $s_{4,3}$ also pay attention to $s_{4,1}, s_{4,2}, s_{4,3}$ of $y_4$.

- **Max**: In 'Max', the state can pay attention to one state of each target token, which has the maximum translating moment. Taking Figure 14(b) as an example, when translating $y_4$, state $s_{4,3}$ can only pay attention to $s_{3,4}$ of $y_3$, as $t_{3,4} = 6$ is the state with the maximum

---

[9]https://www-i6.informatik.rwth-aachen.de/goldAlignment/

[10]Force-decoding: we force the model to generate the ground-truth target token, thereby comparing the translating moments with the ground-truth alignments

[11]For many-to-one alignment, we choose the last source position in the alignment.

translating moments that $s_{4,3}$ ($t_{4,3} = 6$) can pay attention to. Meanwhile, state $s_{4,3}$ pay attention to $s_{4,1}$, $s_{4,2}$, $s_{4,3}$ of $y_4$ as well.

- **Selected**: The most common attention mode in the existing SiMT methods. In 'Selected', the state can pay attention to the selected state used to generate the previous target token. Once the SiMT model determines WRITE (i.e., selects a state), subsequent translations will only depend on this target representation. Taking Figure 14(c) as an example, assuming that $s_{3,3}$ has been selected to generate $y_3$, subsequent $s_{4,2}$, $s_{4,3}$ and $s_{4,4}$ can only focus on the representation of $s_{3,3}$. Note that $s_{4,1}$ cannot focus on $s_{3,3}$ as $t_{3,3} > t_{4,1}$.

**Multiple v.s. Max** The results reported in Table 6 show that 'Multiple' brings 0.65 BLEU improvements compared with 'Max'. The maximum translating moments that 'Multiple' and 'Max' can focus on are the same (in both 'Multiple' and 'Max', state $s_{4,3}$ can pay attention to $s_{4,3}$ with $t_{3,4} = 6$.), but 'Multiple' can also consider those states that start translation earlier, such as $s_{3,1}$, $s_{3,2}$ and $s_{3,3}$. Comprehensively considering the representation of multiple states helps 'Multiple' make more precise judgments and get more accurate state representations (Zhang & Feng, 2021c). Furthermore, in 'Max', the current state always focuses on one previous state with the largest translating moment, regardless of which state is selected to generate the previous target token. For example, even if $s_{4,2}$ is selected to generate $y_3$, state $s_{4,3}$ still only pays attention to $s_{4,3}$ in 'Max', where ignoring the previous selected state $s_{4,2}$ will disturb the dependency of $y_4$ on $y_3$ and thereby affect the translation quality. Therefore, owing to more comprehensive attention to multiple states, the proposed 'Multiple' attention mode achieves better performance.

**Multiple v.s. Selected** 'Selected' is the most commonly used attention mode of the current SiMT method, i.e., once a WRITE action is decided, subsequent target tokens can only pay attention to the representation of this state (Ma et al., 2019; Arivazhagan et al., 2019; Ma et al., 2020). The reason for applying 'Selected' attention mode is that the previous methods can only retain a unique translating moment and corresponding representation for each target token in inference, unlike HMT which can explicitly model multiple translating moments for each target token. Keeping the only target representation of the selected translating moment is susceptible to inaccurate decisions (Zheng et al., 2020b). Assuming that the model selects the state $s_{3,3}$ to generate $y_3$, but this decision is not necessarily completely accurate, it may be more reasonable to use $s_{3,2}$ or $s_{3,4}$ to generate $y_3$. 'Selected' requires subsequent states to only focus on the representation of $s_{3,3}$, which may cause translation errors. 'Multiple' allows the following states to focus on multiple states, including those not selected, and thereby make comprehensive decisions to achieve better results.

In conclusion, 'Max' ignores the previous selected state, 'Selected' only considers the selected state, while the proposed 'Multiple' attention mode pays attention to all previous states and keeps training and inference matching. Therefore, 'Multiple' performs best among these three attention modes.

## D HYPERPARAMETER

Table 7 gives the hyperparameter settings of HMT.

## E NUMERICAL RESULTS WITH MORE METRICS

### E.1 LATENCY METRICS

Besides Average Lagging (AL) (Ma et al., 2019), we additionally use Consecutive Wait (CW) (Gu et al., 2017), Average Proportion (AP) (Cho & Esipova, 2016) and Differentiable Average Lagging (DAL) (Arivazhagan et al., 2019) to evaluate the latency of HMT. We denote the number of waited source tokens before translating $y_i$ as $g_i$ (i.e., the moment to start translating $y_i$), and the calculations of these latency metrics are as follows.

**Consecutive Wait (CW)** (Gu et al., 2017) evaluates the average number of source tokens waited between two target tokens. Given $g_i$, CW is calculated as:

$$\text{CW} = \frac{\sum_{i=1}^{|\mathbf{y}|}(g_i - g_{i-1})}{\sum_{i=1}^{|\mathbf{y}|} \mathbb{1}_{g_i - g_{i-1} > 0}}, \tag{21}$$

Table 7: Hyperparameters of HMT.

| Hyperparameters | En→Vi Transformer-Small | De→En Transformer-Base | De→En Transformer-Big |
|---|---|---|---|
| encoder-layers | 6 | 6 | 6 |
| encoder-attention-heads | 4 | 8 | 16 |
| encoder-embed-dim | 512 | 512 | 1024 |
| encoder-ffn-embed-dim | 1024 | 2048 | 4096 |
| decoder-layers | 6 | 6 | 6 |
| decoder-attention-heads | 4 | 8 | 16 |
| decoder-embed-dim | 512 | 512 | 1024 |
| decoder-ffn-embed-dim | 1024 | 2048 | 4096 |
| dropout | 0.3 | 0.3 | 0.3 |
| optimizer | adam | adam | adam |
| adam-$\beta$ | (0.9, 0.98) | (0.9, 0.98) | (0.9, 0.98) |
| clip-norm | 0 | 0 | 0 |
| lr | 2e-4 | 5e-4 | 5e-4 |
| lr-scheduler | inverse_sqrt | inverse_sqrt | inverse_sqrt |
| warmup-updates | 4000 | 4000 | 4000 |
| warmup-init-lr | 1e-7 | 1e-7 | 1e-7 |
| weight-decay | 0.0001 | 0.0001 | 0.0001 |
| label-smoothing | 0.1 | 0.1 | 0.1 |
| max-tokens | 16000 | 8192×4 | 8192×4 |

where $\mathbb{1}_{g_i - g_{i-1} > 0}$ counts the number of $g_i - g_{i-1} > 0$.

**Average Proportion (AP)** (Cho & Esipova, 2016) evaluates the proportion between the number of received source tokens and the total number of source tokens. Given $g_i$, AP is calculated as:

$$\text{AP} = \frac{1}{|\mathbf{x}| \, |\mathbf{y}|} \sum_{i=1}^{|\mathbf{y}|} g_i. \tag{22}$$

**Average Lagging (AL)** (Ma et al., 2019) evaluates the average number of tokens that target outputs lag behind the source inputs. Given $g_i$, AL is calculated as:

$$\text{AL} = \frac{1}{\tau} \sum_{i=1}^{\tau} g_i - \frac{i-1}{|\mathbf{y}| \, / \, |\mathbf{x}|}, \quad \text{where} \quad \tau = \underset{i}{\text{argmin}} \left( g_i = |\mathbf{x}| \right). \tag{23}$$

**Differentiable Average Lagging (DAL)** (Arivazhagan et al., 2019) is a differentiable version of average lagging. Given $g_i$, DAL is calculated as:

$$\text{DAL} = \frac{1}{|\mathbf{y}|} \sum_{i=1}^{|\mathbf{y}|} g_i' - \frac{i-1}{|\mathbf{x}| \, / \, |\mathbf{y}|}, \quad \text{where} \quad g_i' = \begin{cases} g_i & i = 1 \\ \max \left( g_i, g_{i-1}' + \frac{|\mathbf{x}|}{|\mathbf{y}|} \right) & i > 1 \end{cases}. \tag{24}$$

### E.2 Numerical Results

We adjust $L$ and $K$ in HMT (refer to Eq.(1)) to get the translation quality under different latency. For clarity, we present the numerical results of HMT with the specific setting of hyperparameters $L$ and $K$ in Table 8, Table 9 and Table 10. Note that for comparison, we set $L = -1$ to get the translation quality of HMT under extremely low latency (AL $< 1$, i.e., the outputs lagging the inputs less than 1 token on average) on De→En. When setting $L = -1$, we constrain the translating moment $t_{i,k}$ of all states to be at least 1, i.e., $t_{i,k} = \max \{ \min \{ L + (i - 1) + (k - 1), |\mathbf{x}| \}, 1 \}$. Therefore, all states will start translating after receiving at least 1 source token.

Table 8: Numerical results of HMT on En→Vi with Transformer-Small.

| | | IWSLT15 En→Vi | | Transformer-Small | | |
|---|---|---|---|---|---|---|
| $L$ | $K$ | CW | AP | AL | DAL | BLEU |
| 1 | 2 | 1.15 | 0.64 | 3.10 | 3.72 | 27.99 |
| 2 | 2 | 1.22 | 0.67 | 3.72 | 4.38 | 28.53 |
| 4 | 2 | 1.24 | 0.72 | 4.92 | 5.63 | 28.59 |
| 5 | 4 | 1.53 | 0.76 | 6.34 | 6.86 | 28.78 |
| 6 | 4 | 1.96 | 0.83 | 8.15 | 8.71 | 28.86 |
| 7 | 6 | 2.24 | 0.89 | 9.60 | 10.12 | 28.88 |

Table 9: Numerical results of HMT on De→En with Transformer-Base.

| | | WMT15 De→En | | Transformer-Base | | |
|---|---|---|---|---|---|---|
| $L$ | $K$ | CW | AP | AL | DAL | BLEU |
| -1 | 4 | 1.58 | 0.52 | 0.27 | 2.41 | 22.52 |
| 2 | 4 | 1.78 | 0.59 | 2.20 | 4.53 | 27.60 |
| 3 | 6 | 2.06 | 0.64 | 3.46 | 6.38 | 29.29 |
| 5 | 6 | 1.85 | 0.69 | 4.74 | 6.95 | 30.29 |
| 7 | 6 | 2.03 | 0.74 | 6.43 | 8.35 | 30.90 |
| 9 | 8 | 2.48 | 0.79 | 8.36 | 10.09 | 31.45 |
| 11 | 8 | 3.02 | 0.83 | 10.06 | 11.57 | 31.61 |
| 13 | 8 | 3.73 | 0.86 | 11.80 | 13.08 | 31.71 |

Table 10: Numerical results of HMT on De→En with Transformer-Big.

| | | WMT15 De→En | | Transformer-Big | | |
|---|---|---|---|---|---|---|
| $L$ | $K$ | CW | AP | AL | DAL | BLEU |
| -1 | 4 | 1.65 | 0.52 | 0.06 | 2.43 | 22.70 |
| 2 | 4 | 1.79 | 0.60 | 2.19 | 4.50 | 27.97 |
| 3 | 6 | 2.04 | 0.64 | 3.46 | 6.30 | 29.91 |
| 5 | 6 | 1.88 | 0.69 | 4.85 | 7.07 | 30.85 |
| 7 | 6 | 2.06 | 0.74 | 6.56 | 8.47 | 31.99 |
| 9 | 8 | 2.47 | 0.79 | 8.34 | 10.10 | 32.28 |
| 11 | 8 | 2.98 | 0.83 | 10.12 | 11.58 | 32.46 |
| 13 | 8 | 3.75 | 0.86 | 11.78 | 13.09 | 32.58 |

