# OpenReview forum: "Hidden Markov Transformer for Simultaneous Machine Translation"
_ICLR.cc/2023/Conference — ICLR 2023 notable top 25%_

### Official Review · Reviewer_WoTh · 2022-10-23

**Confidence:** 4
**Correctness:** 3
**Technical Novelty And Significance:** 3
**Empirical Novelty And Significance:** 2
**Recommendation:** 8

**Clarity, Quality, Novelty And Reproducibility:**

* the method is novel and seem to have good quality
* However, as no codes are provided, it's hard to determine if it is reproducible

**Strength And Weaknesses:**

Strength:

* the method is quite novel and experimental results seem strong

Weakness:

* The explanation and presentation of the methodology is complicated and can be reduced.
* More experiments of other language pairs, like french, and low-resource languages (like FLoRes) are needed to confirm the method extendability and generalization.

**Summary Of The Paper:**

The paper proposes a new model called Hidden Markov Transformer (HMT) to tackle the problem of simultaneous machine translation (simt), which tries to translate source to target live as the source is being receive. The model has to decide when to read the source buffer and when to generate or write target token that achieves both low latency and high accuracy.
The method is based on hidden markov chain, which the moments of starting translating are treated as hidden events and it frames the translation results as the corresponding observed events. The model is trained by maximizing the marginal likelihood of the
observed target sequence over multiple possible moments of starting translating
During training, for each target token, is model is trained to output a number of possible states and maximize over which state as the starting state for the current target token.

The results show that the method outperform previous baselines and achieve better latency and accuracy trade-off curve compared to the baselines.

**Summary Of The Review:**

Overall, the method is quite novel and produce good results, even though I suggests to add more language pairs to the experiments.

---

> ### Author Response · Authors · 2022-11-16
> **Thanks for Reviewer WoTh's valuable comments! Here is the response to Reviewer WoTh.**
>
> Thanks for your careful and valuable comments. We have refined the paper following your suggestions. Following, we will respond to your questions in detail.
>
> &nbsp;
>
> Q1: About the explanation and presentation?
>
> A1: Thank you for your suggestion, we will continuously improve the writing of the paper.
>
> &nbsp;
>
> Q2: About language pair in the experiment?
>
> A2: To keep consistent with previous methods, we evaluate HMT on two common SiMT benchmarks, IWSLT15 En-Vi and WMT15 De-En (Base and Big). Specifically, IWSLT15 En-Vi contains a smaller number of language pairs, and WMT15 De-En is recognized as the most challenging language pair in SiMT. The challenge is that German and English have significant differences in linguistic structure, where German is an SOV language (i.e., the verb is at the end of the sentence), and English is an SVO language (i.e., the verb is in the middle of the sentence) (Ma et al., 2019). SiMT from German to English faces the challenge of word order differences, and it is necessary to precisely judge when to start translating (Ma et al., 2019; Zhang & Feng, 2021), so the performance on De-En can evaluate the quality of SiMT policy more effectively. Therefore, we choose these challenging benchmarks to evaluate HMT.

---

### Official Review · Reviewer_4GZM · 2022-10-23

**Confidence:** 4
**Clarity, Quality, Novelty And Reproducibility:** 1. What about BLEU scores in Table 1?…
**Correctness:** 3
**Technical Novelty And Significance:** 3
**Empirical Novelty And Significance:** 3
**Recommendation:** 8

**Strength And Weaknesses:**

Strength:

1. This method explicitly models when to start translating and generating target words;
2. Ablation experiments verified the key operations of the proposed HMT;
3. This study reported that the proposed HMT improves the translation performance on two offline SiMT datasets.

Weaknesses：
1. The training/inference speeds are lower than those of the Wait-k method, which is a disadvantage for SiMT;
2. The interaction and relationship between the proposed HMT and the read/write policy were not clearly introduced;
3. Too many hyperparameters (i.e., L、K、Lamda1, and Lamda2) are not conducive to the optimization of this method on other language pairs;
4. There lack of necessary experiment to verify what the proposed HMT indeed capture compared to the baseline wait-k.

**Summary Of The Paper:**

This paper proposes a hidden Markov Transformer (HMT) for simultaneous machine translation (SiMT). The proposed HTM is able to simulate when to start translating and to generate the target token. The results of the proposed method are reported on two offline simultaneous translation datasets.




**Summary Of The Review:**

This is an extension of the existing wait-k and causes high latency.


The author's response addressed my concerns, and thank you very much. I have raised the score to 8

---

> ### Author Response · Authors · 2022-11-16
> **Thanks for Reviewer 4GZM's valuable comments! Here is the response to Reviewer 4GZM (PART II).**
>
> Q6: The attention on the states adopts multiple modes. Why the multiple modes are the best in terms of results in Table 6?
>
> A6: In 'Multiple', the state can pay attention to multiple states of previous target tokens. While in 'Max', the state can only pay attention to one state of each target token, which has the maximum translating moment. In 'Selected', the state can only pay attention to the selected state used to generate the previous target token. Taking the HMT with $L=1$ and $K=4$ for example, the translating moments of states for $y_{3}$ and $y_{4}$ are $t_{3}=(3,4,5,6)$ and $t_{4}=(4,5,6,7)$, respectively. The main reasons why 'Multiple' outperforms 'Max' and 'Selected' include:
>
> - **'Max' will ignore the previous selected state**: In 'Max', the current state always focuses on one of the previous states with the largest translating moment, regardless of which state is selected to generate the previous target token. For example, even if $s_{4,2}$ is selected to generate $y_{3}$, state $s_{4,3}$ still only pays attention to $s_{4,3}$ in 'Max', where ignoring the previous selected state $s_{4,2}$ will disturb the dependency of $y_{4}$ on $y_{3}$ and thereby affect the translation quality. Therefore, owing to more comprehensive attention to multiple states, 'Multiple' achieves better performance.
> - **'Selected' can only focus on the selected state**: In 'Selected', the state can only pay attention to the selected state used to generate the previous target token. Assuming that the model selects the state $s_{3,3}$ to generate $y_3$, but this decision is not necessarily completely accurate, it may be reasonable to use $s_{3,2}$ or $s_{3, 4}$ to generate $y_3$. 'Selected' requires subsequent states to only focus on representations of $s_{3,3}$, which may cause translation errors. 'Multiple' allows the following states to focus on all states, including those not selected, and thereby make comprehensive decisions to achieve better results.
>
> - **'Multiple' focuses on more states**: When calculating the representation of state $s_{4,3}$, 'Max' will focus on $s_{3,4}$ with the maximum translating moments and 'Selected' will only focus on the selected state, such as $s_{3,3}$. While 'Multiple' can focus on more states $s_{3,1}$, $s_{3,2}$, $s_{3,3}$ and $s_{3,4}$. Therefore, comprehensively considering the representation of multiple states helps 'Multiple' make more precise judgments and get more accurate state representations (Zhang & Feng, 2021), thereby achieving better performance.
>
> Thanks for your question. In the new version, to compare the attention modes in HMT more clearly, we give a schematic diagram of the various attention modes ('Multiple', 'Max' and 'Selected') in Appendix C.7, and further explain the advantages of 'Multiple'.
>
> &nbsp;
>
> Q7: Why does the parameter amount not increase?
>
> A7: Thanks for pointing this out. HMT has only 1024 more parameters than full-sentence MT, so Table 1 cannot show it. In the new version, we have put more detailed numerical results of Table 1 into Appendix C.6.
>
> &nbsp;
>
> Q8: About source code?
>
> A8: We will release the source code in the final version.
>
> &nbsp;
>
> If our responses answer your questions well and reassure your concerns, we would appreciate if you could reassess our work.

---

> ### Author Response · Authors · 2022-11-16
> **Thanks for Reviewer 4GZM's valuable comments! Here is the response to Reviewer 4GZM (PART I).**
>
> Thanks for your careful and valuable comments. We have refined the paper following your suggestions. Following, we will respond to your questions in detail.
>
> &nbsp;
>
> Q1: The training/inference speeds are lower than those of the Wait-k method, which is a disadvantage for SiMT.
>
> A1: Indeed, the training and inference speeds of HMT are slightly lower than the wait-k policy. However, for training, given the obvious performance improvements, we argue that the slightly slower training speed than the fixed method is completely acceptable. For inference, if the wait-k policy wants to achieve the same translation quality as HMT, it needs to wait for about 2 more tokens (As shown in Figure 4, under the same BLEU score, AL of wait-k is about 2 tokens higher than HMT). Compared to these extra waits, the slightly slower inference speed is almost negligible in inference. Besides, compared with other SOTA adaptive methods, HMT has obvious advantages in both training and inference speeds.
>
> In the new version, we gave a more detailed introduction to speed in Appendix C.6.
>
> &nbsp;
>
> Q2: The interaction and relationship between the proposed HMT and the read/write policy were not clearly introduced.
>
> A2: Thank you for your question. The introduction of HMT inference in Sec.3.3 includes the relationship between HMT and read/write policy, but I am sorry that this introduction may not be clear. For the relationship between HMT and read/write policy, selecting a state in the HMT corresponds to performing WRITE action, while not selecting the current state corresponds to performing READ action.
>
> Following your suggestion, we have refined Sec.3.3 and Algorithm 1 in the new version, in which we emphasize and give a clear relationship between the proposed HMT and the read/write policy.
>
> &nbsp;
>
> Q3: Too many hyperparameters (i.e., $L$、$K$、$\lambda_{latency}$ and $\lambda_{state}$) are not conducive to the optimization of this method on other language pairs?
>
> A3: Actually, the only hyperparameter that HMT needs to search is the state number $K$.
>
> - For $\lambda_{latency}$ and $\lambda_{state}$ in the loss function: HMT is not sensitive to them and we empirically set them to 1 on all language pairs in our experiments, so we do not need to search them overmuch.
> - For $L$: It controls the overall latency of SiMT model to get the translation quality under different latency (i.e., getting those points in Figure 4). $L$ is necessary for SiMT evaluation and you can set L to any value to get various latency. All existing SiMT methods require a similar parameter to control the overall latency.
> - For $K$: different $K$ settings will slightly affect the translation quality, so we give a general guide on how to choose $K$ in Sec. 5.2.
>
> Therefore, HMT does not require excessive hyperparameter searching when applied to different language pairs.
>
> &nbsp;
>
> Q4: Lack of necessary experiment to verify what the proposed HMT indeed captures compared to the baseline wait-k.
>
> A4: Following your suggestion, in the new version, we add an analytical experiment in Appendix C.5 to verify what the proposed HMT captures compared to the baseline wait-k. The results show that the better performance of HMT is attributed to the more precise SiMT policy brought by HMT. Specifically, under the same latency, HMT can receive more aligned source tokens than wait-k policy before starting translating (i.e., HMT learns to start translating at more suitable moments), thereby achieving higher translation quality.
>
> &nbsp;
>
> Q5: What about BLEU scores in Table 1?
>
> A5: Due to space limitations, we do not report BLEU scores in Table 1. Following your suggestion, in the new version, we report more detailed numerical results of Table 1 in Appendix C.6. Results show that HMT can achieve higher BLEU scores under similar latency.

---

### Official Review · Reviewer_NSCn · 2022-10-24

**Confidence:** 5
**Correctness:** 4
**Technical Novelty And Significance:** 2
**Empirical Novelty And Significance:** 4
**Recommendation:** 6

**Clarity, Quality, Novelty And Reproducibility:**

This article did not release the code but used open-source deep-learning frameworks and public datasets. Therefore, I believe this work can be reproduced with some effort.

**Strength And Weaknesses:**

Strength:
1. The proposed HMT approach gained improvement over the baseline systems in terms of BLEU scores and latency.
2. Experiments have proved the effectiveness of the method, and there are very reasonable expanded analyses.

Weaknesses:
The average pooling result on the hidden states of the received source is only a summary of the past state. And this model didn't use an explicit alignment method. Therefore, the proposed approach is more like a pruning method to cut low probability branches of the READ/WRITE path.  This doesn't quite match what the paper claims that explicitly models multiple possible moments of starting translating in both training and inference.

Questions:
1. How to evaluate the proposed method to make better read/write decisions than the other SNMT?
2. If there is higher confidence in the later state, will the performance of the model be affected by generating too early?
3. Can you provide an evaluation of the quality, clarity, and originality of the work?

**Summary Of The Paper:**

This paper used a Hidden Markov Transformer(HMT) to improve the READ/WRITE actions in simultaneous neural machine translation. Specifically, this model treats the moments of starting translating as hidden events and the target sequence as the corresponding observed events. The HMT model judges whether to select each state from low to high latency. The authors hoped that the HMT model effectively learns when to start translating under the supervision of the observed target sequence. The proposed model was evaluated on two language pairs and gained improvement over the baseline systems in terms of BLEU scores and latency.

**Summary Of The Review:**

This paper provides a new perspective on simultaneous neural machine translation, and sufficient experiments prove the effectiveness of the method. Compared with several strong baseline systems, the proposed method achieves state-of-the-art performance.

---

> ### Author Response · Authors · 2022-11-16
> **Thanks for Reviewer NSCn's valuable comments! Here is the response to Reviewer NSCn (PART II).**
>
> Q4: If there is higher confidence in the later state, will the performance of the model be affected by generating too early?
>
> A4: We also took into account what you mentioned, so we verified the performance of different confidence thresholds in Appendix C.2. As shown in Figure 9, when the confidence threshold $\delta=0.8$, HMT will wait until reaching the later state with a higher confidence, but the improvement in translation quality is not obvious. This is mainly because the introduced state loss encourages all states to generate correct target tokens. Therefore, the performance of HMT will not be obviously affected when setting $\delta=0.5$.
>
> &nbsp;
>
> Q5: Can you provide an evaluation of the quality, clarity, and originality of the work?
>
> A5: The proposed HMT is completely different from previous methods. As far as we know, HMT is the first approach to integrate 'learning when to start translating' and 'learning translation' into a hidden Markov model, which successfully models the relationship between these two key issues in SiMT. Experiment results on 2 benchmarks show that HMT achieves state-of-the-art performance, and extensive analyses demonstrate the effectiveness of HMT. Besides, detailed case studies show how HMT works and its interpretability. Overall, we argue that HMT provides a distinct SiMT policy, which is novel and interesting.
>
> &nbsp;
>
> Q6: About source code?
>
> A6: We will release the source code in the final version.
>
> &nbsp;
>
> If our responses answer your questions well and reassure your concerns, we would appreciate if you could reassess our work.

---

> ### Author Response · Authors · 2022-11-16
> **Thanks for Reviewer NSCn's valuable comments! Here is the response to Reviewer NSCn (PART I).**
>
> Thanks for your careful and valuable comments. We have refined the paper following your suggestions. Following, we will respond to your questions in detail.
>
> &nbsp;
>
> Q1: Why use the average pooling result of the received source hidden states to decide READ/WRITE?
>
> A1: As you mentioned, we use the average pooling result of all received source hidden states to predict the confidence of selecting the state, which is an innovation of our method. A good SiMT policy should start translating after receiving the related source token, so our motivation is to learn when to start translating based on the received source content.
>
> The previous policy, such as MMA, determines READ/WRITE based on the target token and the last source token, which judges whether the last source token and the target token are related. This approach is difficult to deal with those non-monotonic alignments because it only models the last source token rather than the entire received source content. For example, if the target token is aligned with the previous source token instead of the last source token, it is difficult for MMA to accurately decide READ/WRITE. In HMT, using the average pooling result of the received source hidden states can consider all received source content and accordingly judge whether the received source content is related to the translation of the current target token, enabling more precise READ/WRITE decisions.
>
> In the new version, we added an evaluation of policy quality in Appendix C.5. The results show that HMT can start translating after receiving more aligned source tokens under the same latency, demonstrating that HMT can make more precise decisions on when to start translating.
>
>
> &nbsp;
>
>
> Q2: HMT doesn't quite match what the paper claims that explicitly models multiple possible moments of starting translating.
>
> A2: The architecture of the HMT indeed explicitly models the multiple possible moments of starting translating.
>
> - First of all, in SiMT, the moments of starting translating correspond to which source tokens are received before starting translating. (**translating moment $\Leftrightarrow$ received source tokens before starting translating**)
>
> - To model different translating moments, HMT explicitly introduces multiple states and restricts each state to receive a different number of source tokens. (**translating moment $\Leftrightarrow$ a state, multiple translating moments $\Leftrightarrow$ multiple states**)
> - Meanwhile, HMT also explicitly models the dependencies between multiple possible moments of starting translating via the attention between states. (**dependencies between multiple translating moments $\Leftrightarrow$ self-attention between states**)
> - Then, when predicting the state confidence, HMT also uses all the current received source tokens, i.e., which source tokens are received before starting translating.  (**whether the current translation moment is confident $\Leftrightarrow$ predict confidence based on the received source tokens (pooling results)**)
>
> Therefore, the motivation and method of HMT are self-consistent in terms of definition, modeling and confidence prediction.
>
> &nbsp;
>
> Q3: How to evaluate the proposed method to make better read/write decisions than the other SNMT?
>
> A3: Thanks for your question. In the new version, we added an evaluation of policy (i.e., READ/WRITE decisions) quality in Appendix C.5. Following Zhang & Feng (2022a), we calculate the proportion of the ground-truth aligned source tokens received before translating in various SiMT policies, where better policy can receive more aligned token under the same latency. As shown in Figure 13, compared with previous policies, HMT receives more aligned source tokens before translating under the same latency, which demonstrates that HMT can make more precise decisions on when to start translating.

---

### Official Review · Reviewer_C1gX · 2022-10-25

**Confidence:** 4
**Correctness:** 3
**Technical Novelty And Significance:** 3
**Empirical Novelty And Significance:** 3
**Recommendation:** 8

**Clarity, Quality, Novelty And Reproducibility:**

The paper is basically clear, except for several previously mentioned questions. The method is interesting and intuitive for the proposed problem. The experiments are extensive and solid.

**Strength And Weaknesses:**

Reasons to accept:
- The methods produces latent inputs for the translation to boost latency, which is interesting and promising.
- The conducted experiments are extensive, providing in-depth analysis on how the proposed method works.


Questions:
1. Is the encoder in Full-Sentence MT bi-directional or uni-directional? If uni-directional, an alternative paper should be cited instead the original Transformer paper.
2. Why HMT even outperforms Full-Sentence MT in De->En? Also in Section 5.3, I don't understand why Multiple is better than Max, where the latter always provides more information than the former when generating. Could authors please provide more explanations?
3. What is the system overhead (e.g., memory) of HMT compared to baselines? Since HMT produces K states for each timestep, and self attentions are conducted on these multiple states, the memory requirements for SA would be K^2 compared to vanilla models. This could be a concern for edge devices.
Missing reference:
Liu D, Du M, Li X, et al. Cross attention augmented transducer networks for simultaneous translation[C]//Proceedings of the 2021 Conference on Empirical Methods in Natural Language Processing. 2021: 39-55.

**Summary Of The Paper:**

This paper focuses on the core issue of simultaneous machine translation: time to start translation. The paper proposes HMT inspired by HMM, where the next token generation is regarded as the hidden variables, and target tokens as the observable variables. At each time step, K indicates available length of source for translation, and each state predicts the next token to be yield. During training, the marginal probability of generating the correct token is maximized. During inference, the first state with over 0.5 confidence is selected as the generated token.

Experiments conducted on two SiMT benchmarks validate the effectiveness of HMT. Extensive experiments are also conducted to promote better understanding of how HMT works.

**Summary Of The Review:**

This paper addresses the core problem of simultaneous machine translation: when to start translate. The paper propose HMT as a solution. The experiments demonstrates the effectiveness of the proposed method.

---

> ### Author Response · Authors · 2022-11-16
> **Thanks for Reviewer C1gX's valuable comments! Here is the response to Reviewer C1gX.**
>
> Thanks for your careful and valuable comments. We have refined the paper following your suggestions. Following, we will respond to your questions in detail.
>
> &nbsp;
>
> Q1: Is the encoder in Full-Sentence MT bi-directional or uni-directional?
>
> A1: Indeed, the encoder of full-sentence MT is uni-directional. Thanks for your suggestion, we cite an alternative paper and highlight this in the new version.
>
> &nbsp;
>
> Q2: Why HMT even outperforms Full-Sentence MT in De->En?
>
> A2: On De->En, full-sentence MT achieves 31.60 BLEU, and HMT achieves 31.71 BLEU when lagging 10 source tokens, improving about 0.1 BLEU compared with full-sentence MT. The improvement mainly comes from explicitly modeling multiple states and applying self-attention between states can slightly improve the translation ability of the model, about 0.2 BLEU.
>
> Besides, the superiority of HMT policy also helps it achieve similar performance to full-sentence MT. In the new version, we added an evaluation of the policy quality of HMT in Appendix C.5. When lagging about 10 tokens, HMT can receive almost 99.2% of the aligned source tokens before translating, so its translation performance can be comparable to full-sentence MT.
>
> &nbsp;
>
> Q3: Why Multiple is better than Max?
>
> A3: In 'Multiple', the state can pay attention to multiple states of previous target tokens. While in 'Max', the state can only pay attention to one state of each target token, which has the maximum translating moment. Taking the HMT with $L=1$ and $K=4$ for example,  the translating moments of states for $y_{3}$ and $y_{4}$ are $t_{3}=(3,4,5,6)$ and $t_{4}=(4,5,6,7)$, respectively. The main reasons why 'Multiple' outperforms 'Max' include:
>
> - **'Multiple' focuses on more states**: When calculating the representation of state $s_{4,3}$, 'Max' will focus on $s_{3,4}$ with the maximum translating moments, while 'Multiple' can focus on $s_{3,1}$, $s_{3,2}$, $s_{3,3}$ and $s_{3,4}$. Therefore, comprehensively considering the representation of multiple states helps 'Multiple' make more precise judgments and get more accurate state representations (Zhang & Feng, 2021), thereby achieving better performance.
>
> - **'Max' will ignore the previous selected state**: Furthermore, in 'Max', the current state always focuses on one of the previous states with the largest translating moment, regardless of which state is selected to generate the previous target token. For example, even if $s_{4,2}$ is selected to generate $y_{3}$, state $s_{4,3}$ still only pays attention to $s_{4,3}$ in 'Max', where ignoring the previous selected state $s_{4,2}$ will disturb the dependency of $y_{4}$ on $y_{3}$ and thereby affect the translation quality. Therefore, owing to more comprehensive attention, 'Multiple' achieves better performance.
>
> Thanks for your question. In the new version, to compare the attention modes in HMT more clearly, we give a schematic diagram of the various attention modes ('Multiple', 'Max' and 'Selected') in Appendix C.7, and further explain the advantages of 'Multiple'.
>
> &nbsp;
>
> Q4: What is the system overhead (e.g., memory) of HMT compared to baselines?
>
> A4: Thanks for your concern about system overhead. HMT upsamples the decoder input K times during training, where memory requirements for decoder self-attention will be $K^2$ times compared to vanilla models. However, compared to other powerful adaptive methods such as MMA and GSiMT, the memory requirements of HMT are minimal. MMA needs to loop the decoder cross-attention about $n$ times, where $n$ is the sequence length, and complex computations further generate many intermediate variables (about 3 matrices in each loop). GSiMT needs to loop the Transformer $n^2$ times to obtain the representation under all READ/WRITE combinations. Therefore, compared to these adaptive methods, the memory overhead of HMT is minimal, and parallelizable attention operations do not generate additional intermediate variables.
>
> &nbsp;
>
> Q5: About missing reference?
>
> A5: Thanks for your reminder, we've added the reference (Liu et al., 2021) to the related work in the new version.

---

### Author Response · Authors · 2022-11-16
**Paper has been revised according to the reviewers' valuable suggestions!**

Thanks for all reviewers' careful comments and efforts on our work. In the new version, we have refined the paper following your suggestions, mainly including:

- An evaluation of HMT policy quality is added in Appendix C.5 to demonstrate the superiority of HMT in deciding when to start translation (i.e., READ/WRITE decisions).
- Detailed results on parameters and speeds of HMT are added in Appendix C.6.
- An in-depth explanation of why the 'Multiple' attention mode performs better is added in Appendix C.7.
- Some writing and presentation are improved.

---

### Decision · Program_Chairs · 2023-01-20

**Decision:**

Accept: notable-top-25%

**Justification For Why Not Higher Score:**

The Markov-based SNMT is a specific NLP task. I don't think this paper will broadly interest a large amount of NLP audiences.

**Justification For Why Not Lower Score:**

It is a clear acceptance.

**Metareview: Summary, Strengths And Weaknesses:**

This paper proposed a Markov-based Transformer model for simultaneous NMT (SNMT). In all, this paper is well-written. All the reviewers agreed that this paper is interesting and applying the Markov method to SNMT is reasonable. It is a clear acceptance.

**Note From Pc:**

if the above contains the word "oral" or "spotlight" please see: "oral" presentation means -> notable-top-5% and "spotlight" means -> notable-top-25%. As stated in our emails, we are disassociating presentation type from AC recommendations

**Summary Of Ac-Reviewer Meeting:**

n/a